**Citation:** *Molecular Systems Biology* 9:682
www.molecularsystemsbiology.com

# A yeast one-hybrid and microfluidics-based pipeline to map mammalian gene regulatory networks

Carine Gubelmann, Sebastian M Waszak, Alina Isakova, Wiebke Holcombe, Korneel Hens, Antonina Iagovitina, Jean-Daniel Feuz, Sunil K Raghav, Jovan Simicevic and Bart Deplancke*

Institute of Bioengineering, School of Life Sciences, Ecole Polytechnique Fédérale de Lausanne (EPFL), Lausanne, Switzerland
* Corresponding author. Laboratory of Systems Biology and Genetics (LSBG), EPFL-SV-IBI-LSBG, Station 19, 1015 Lausanne, Switzerland. Tel.: + 41 21 693 18 21; Fax: + 41 21 693 09 80; E-mail: bart.deplancke@epfl.ch

**The comprehensive mapping of gene promoters and enhancers has significantly improved our understanding of how the mammalian regulatory genome is organized. An important challenge is to elucidate how these regulatory elements contribute to gene expression by identifying their *trans*-regulatory inputs. Here, we present the generation of a mouse-specific transcription factor (TF) open-reading frame clone library and its implementation in yeast one-hybrid assays to enable large-scale protein–DNA interaction detection with mouse regulatory elements. Once specific interactions are identified, we then use a microfluidics-based method to validate and precisely map them within the respective DNA sequences. Using well-described regulatory elements as well as orphan enhancers, we show that this cross-platform pipeline characterizes known and uncovers many novel TF–DNA interactions. In addition, we provide evidence that several of these novel interactions are relevant *in vivo* and aid in elucidating the regulatory architecture of enhancers.**
*Molecular Systems Biology* **9**: 682; published online 6 August 2013; doi:10.1038/msb.2013.38
*Subject Categories:* chromatin & transcription
*Keywords:* gene regulatory networks; microfluidics; mouse open-reading frame (ORF) clone collection; transcription factor; yeast one-hybrid

## Introduction

Since the original discovery of a cellular enhancer (Banerji *et al*, 1983), the mouse has been an excellent model organism for studying metazoan gene regulation, especially mammalian-specific complex biological processes, due to the availability of tissues and ease of genomic manipulation (Paigen, 2003; Stamatoyannopoulos *et al*, 2012). An example of the latter is the ability to characterize the transcriptional activity of putative gene regulatory elements *in vivo*. In recent years, such elements have been identified in increasing numbers and at a genome-wide level through computational approaches (Siepel *et al*, 2005; Hallikas *et al*, 2006; Birney *et al*, 2007; Ferretti *et al*, 2007), co-regulator or histone modification mapping (Birney *et al*, 2007; Heintzman *et al*, 2009; Visel *et al*, 2009; Shen *et al*, 2012; Raghav *et al*, 2012), and large-scale initiatives such as the mouse ENCODE will further expand this element catalog (Stamatoyannopoulos *et al*, 2012). Together with studies revealing the genome-wide location of gene promoters in the mouse (Carninci *et al*, 2006), these efforts have already led to a better knowledge of the regulatory architecture of the mammalian genome. To understand how regulatory elements contribute to gene expression, we now need to elucidate the transcription factors (TFs) that bind to them (Nam *et al*, 2010). The most convenient strategy to identify *trans*-regulatory

inputs has typically been the computational prediction of TF binding sites within the regulatory elements of interest (Wasserman and Sandelin, 2004). Specifically, elements are scanned for potential TF binding sites using positional weight matrices (PWMs), which represent the DNA binding specificities of TFs (Stormo and Zhao, 2010). A recent comparison of such computational scanning tools revealed that the best performing ones achieved high prediction accuracy, although non-negligible discrepancies between experimentally- and *in silico*-derived data remained (Weirauch *et al*, 2013). Consequently, downstream experimental validation will always be required to conclusively identify implicated TFs. In addition, despite valuable efforts (Berger *et al*, 2008; Noyes *et al*, 2008; Jolma *et al*, 2013), the binding specificities of the majority of mammalian TFs have still not been characterized. Identifying TF–DNA interactions based on PWM models alone will therefore leave a significant portion of the regulatory space unprobed.

An alternative approach to link TFs with regulatory elements involves the use of protein-centered protein–DNA interaction detection methods such as chromatin immunoprecipitation (ChIP) coupled to high-throughput sequencing (ChIP-seq) (Johnson *et al*, 2007; Robertson *et al*, 2007). However, ChIP-seq has a low throughput on the TF side, is

restricted by the availability of TF-specific antibodies, and often fails when targeted TFs are, for example, expressed in a small time window or in a limited number of cells (Deplancke, 2009; Farnham, 2009). This explains why the transcriptional activity for only a moderate number of regulatory elements has so far been linked to the binding of specific TFs using ChIP-seq. To alleviate some of the issues associated with motif-based or protein-centered TF–regulatory element interaction identification, we and others have previously developed DNA-centered yeast one-hybrid (Y1H)-based methods that allow the large-scale detection of human (Reece-Hoyes *et al*, 2011a), *Drosophila melanogaster* (Hens *et al*, 2011), and *Caenorhabditis elegans* (Deplancke *et al*, 2004; Reece-Hoyes *et al*, 2011b) proteins that bind to DNA elements of interest. For each of these species, comprehensive TF open-reading frame (ORF) clone libraries were generated so that in principle any regulatory element can be screened against the majority of TFs of the respective species in a PWM-independent manner. Surprisingly, however, there has so far not been a concerted effort to build a similar toolkit for mouse (*Mus musculus*), 'despite' significant on-going efforts such as the mouse ENCODE (Stamatoyannopoulos *et al*, 2012).

With the aim of characterizing predicted mammalian regulatory elements, especially those that exhibit transcriptional activity *in vivo*, we have generated a mouse-specific TF ORF library and employed it in Y1H assays, enabling large-scale protein-DNA interaction identification for the mouse. Building on pioneering work in *D. melanogaster* (Hens *et al*, 2011), we also show how our Y1H system can be coupled to a microfluidics-based protein–DNA interaction mapping assay, *M*ITOMI-based *A*nalysis of *R*egulatory *E*lements (MARE). MARE enables the simultaneous monitoring of $>700$ protein–DNA interactions, rendering this technology ideal to systematically scan relatively long ($\sim 1$ kb) regulatory elements for DNA occupancy of specific TFs. MARE is therefore complementary to the Y1H assay because it allows the validation of Y1H-detected interactions and hence the filtering of potential false positives (Hens *et al*, 2011). In addition, MARE enables fine-grained mapping of detected interactions, and thus offers far greater individual TF binding site resolution than Y1H. Our Y1H and MARE methods therefore constitute a powerful cross-platform pipeline to experimentally characterize the mammalian regulatory elements of interest for interacting TFs at unprecedented throughput and resolution.

## Results

### A Gateway-compatible mouse TF ORF clone resource

We have generated a comprehensive, Gateway-compatible mouse (*Mus musculus*) TF ORF clone resource. On the basis of manual curation and previously published bioinformatics analyses (Kummerfeld and Teichmann, 2006), we identified 1576 TF-coding genes in the mouse genome (Materials and methods). We retrieved the cDNA templates for 1260 TFs ($\sim 80\%$) and could derive ORFs for 1014 ($\sim 64\%$) of them using PCR and Gateway cloning. Using a high-throughput sequencing-based sequence identification approach (Massouras *et al*, 2010), we were able to verify that

750 ($\sim 48\%$) of these TF ORF clones contain the respective full-length reference ORF sequence (Figure 1A; Supplementary Table S1). While large-scale efforts have previously been undertaken to clone all mouse cDNAs (Fantom Consortium, 2005), the presented work represents to our knowledge the first systematic effort of generating a mouse-specific, Gateway-compatible, and fully sequence-verified TF ORF library.

### Benchmarking of the mouse yeast one-hybrid assay

To perform large-scale Y1H assays, we considered two different, but complementary screening formats (Figure 1B). The first is the diploid yeast-based screening of DNA elements in which yeast strains containing one DNA bait linked to a reporter gene (e.g., *HIS3*) are mated with an array of TF-expressing (i.e., prey) yeast strains. This strategy has the advantage of being cost- and throughput-effective and can be scaled up to accommodate a great number of prey proteins (Vermeirssen *et al*, 2007; Reece-Hoyes *et al*, 2011b). However, it is relatively labor-intensive due to the many intermediate steps between the mating and final selection phase, and thus difficult to completely automate. We therefore also considered a yeast-based matrix screening format, in which TFs are individually transformed into haploid, DNA element-containing yeast strains. While in practice more labor and cost intensive than mating, we have recently automated and miniaturized this matrix-based screening process, now enabling the hands-off screening of DNA elements versus a large array of TFs (Materials and methods; Hens *et al*, 2011).

To compare and validate these two strategies for mouse protein–DNA interaction detection studies, we selected 10 DNA baits. Eight of these are well-characterized regulatory elements for which several interactors have been identified using different types of protein–DNA interaction detection methods (Supplementary Table S2; Materials and methods). These interactors can be used as positive controls, allowing us to assess the performance of our Y1H platform. To illustrate the discovery potential of our Y1H set-up, we also included two randomly selected orphan enhancers that were shown to exhibit regulatory activity in mouse transgenic assays (Visel *et al*, 2009). Three of the ten baits, including one enhancer, exhibited high self-activation activity and were eliminated from further analyses. The remaining elements were screened versus a mouse-specific array of 745 TFs, which was generated by transferring all 750 sequence-verified TF ORFs to Y1H-compatible destination vectors (Figure 1A; Supplementary Table S1). We identified 40 protein–DNA interactions involving 25 distinct TFs and 7 DNA baits. The results of two independent screens involving these DNA baits are presented in Table I, Figure 2, and Supplementary Figures S1–S7.

On the basis of these data, four conclusions can be drawn: *first*, direct transformation tends to yield a greater number of positive interactions per screen compared with the mating procedure (Figure 2; Table I, last row; Supplementary Table S3), consistent with previous results in *C. elegans* (Vermeirssen *et al*, 2007). However, mating results tend to be more reproducible as most of the interactions found in the first mating screen were also detected in the second (Table I, last

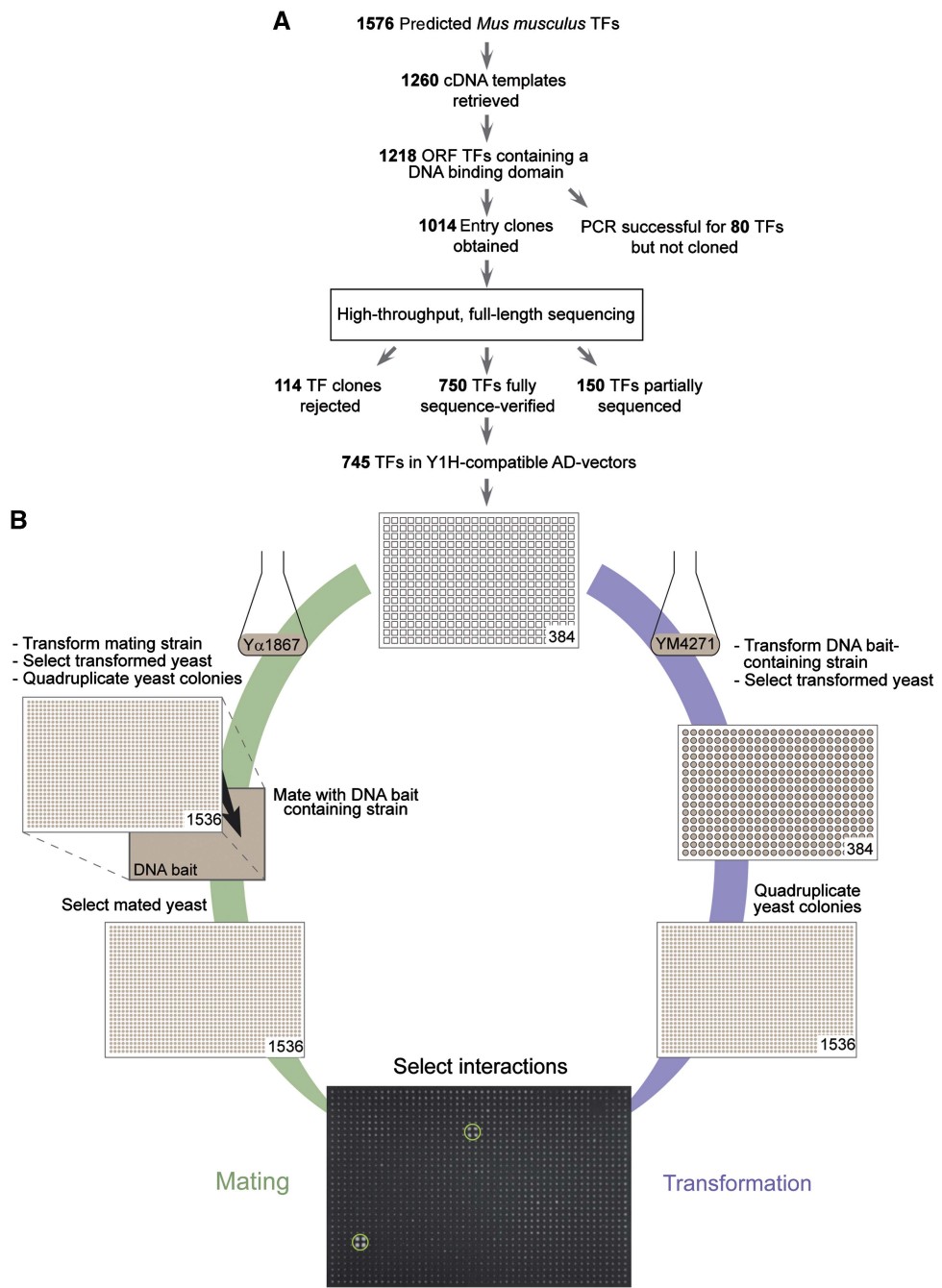

**Figure 1** Mouse Y1H pipeline. (**A**) Workflow to retrieve the mouse TF open-reading frame (ORF) clone resource. We predict that the mouse genome contains 1576 TF-coding genes (Materials and methods). We were able to retrieve cDNA templates for 1260 of these. To initiate the cloning of the corresponding ORFs, we first applied a strong selection scheme choosing the longest ORF per TF containing the respective DBD. After filtering, 1218 TF cDNA templates were retained. So far, 1014 TF ORFs have been cloned. The corresponding entry clones were sequence verified using high-throughput sequencing. In all, 150 and 750 TFs were, respectively, partially or fully sequence verified. (**B**) The mouse Y1H screening pipeline using either the mating (diploid Y1H) or the direct transformation (haploid Y1H) protocol. The 745 TFs in Y1H-compatible AD-vectors are transformed into the yeast DNA bait strain or into the mating yeast strain for the transformation and the mating assays, respectively; the colonies are then selected and four replicates are created by using a colony-pinning robot that transfers the 384 individually transformed yeast strains and spots them in a square pattern. In the mating protocol, the mating yeast strains containing the respective TFs are mated with the yeast DNA bait strain and then selected to create diploid yeast. Finally, in both protocols, TF–DNA bait interactions are identified based on growth on a selective 3-amino-1,2,4-triazole (3-AT)-containing yeast plate. The green circles indicate positive protein–DNA interactions.

row, columns three and seven). One possible reason for the observed difference in interaction reproducibility between the two Y1H screening procedures may be that TFs are re-introduced into fresh yeast strains in the transformation procedure, whereas the same TF-containing yeast strains are typically used to repeat the mating screen (Figure 1B; Materials and methods). *Second*, among the reproducibly detected interactions found by either mating or transformation, only

**Table I** Summary of the transformation and mating-based Y1H screening data

| Baits | Screens | | | | | | | | Consensus (i.e., reproducibly found using both screening procedures) |
|---|---|---|---|---|---|---|---|---|---|
| | Transformation | | | | Mating | | | | |
| | First screen | Second screen | Overlap | Known interactors | First screen | Second screen | Overlap | Known interactors | |
| *Rhs7-2* element | 10 | 13 | 6/10 | 0/2 | 7 | 7 | 7/7 | 0/2 | 3/10 |
| *M. musculus* control region of beta-globin locus (*Mlcrhs4* element) | 9 | 15 | 9/9 | 0/1 | 3 | 3 | 3/3 | 0/1 | 3/9 |
| *Ptgs2* promoter | 6 | 6 | 1/6 | 1/9 | 2 | 2 | 1/2 | 1/9 | 1/1 |
| *Pou5f1* promoter | 6 | 11 | 3/6 | 3/7 | 9 | 9 | 8/9 | 4/7 | 3/8 |
| *Mcts2-Id1* enhancer element | 6 | 9 | 3/6 | — | 2 | 3 | 2/2 | — | 2/3 |
| *Fos* promoter | 14 | 11 | 3/14 | 0/4 | 3 | 3 | 2/3 | 0/4 | 0/5 |
| *Mmp9* promoter | 5 | 7 | 3/5 | 1/5 | 2 | 2 | 2/2 | 2/5 | 1/4 |
| Total interactors (percentage) | 56 | 72 | 28/56 (50%) | 5/28 (18%) | 28 | 29 | 25/28 (89%) | 7/28 (25%) | 13/40 (33%) |

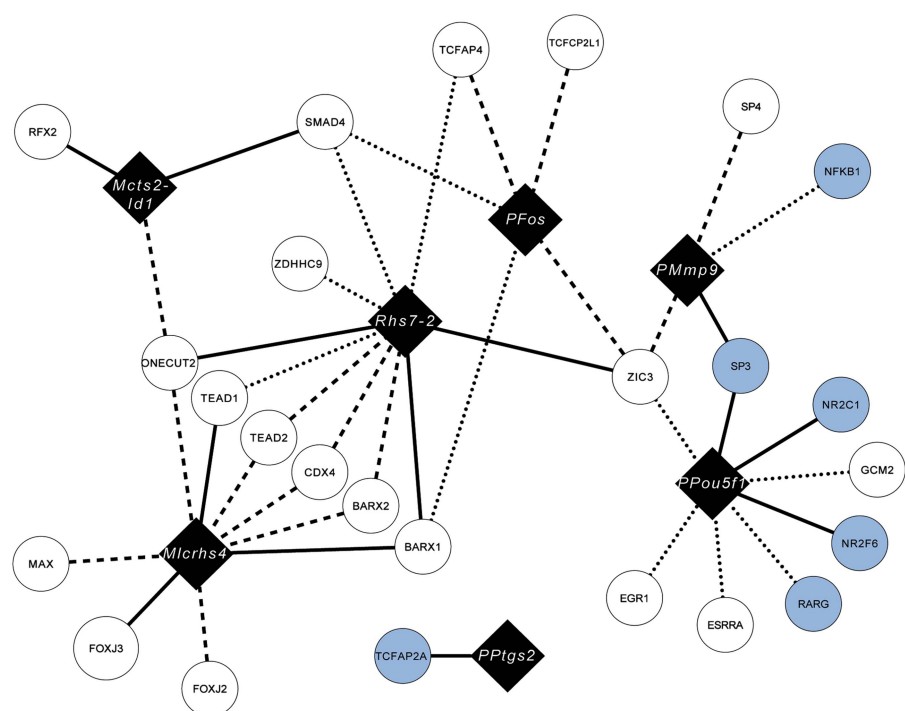

**Figure 2** Network representation of the detected Y1H interactions. This network generated by Cytoscape (Smoot *et al*, 2011) shows the identified interactions between the tested DNA baits (black diamonds) and the Y1H-detected TF interactors (circles). Consensus interactions are represented by solid lines. Interactions reproducibly found by either transformation or mating are represented by dashed- or dotted lines, respectively. Interactors reported in the literature (Supplementary Table S2) are highlighted in light blue. 'P' refers to promoter.

13 (33%) out of 40 protein–DNA interactions were in common revealing extensive interassay variability consistent with what has been observed for yeast two-hybrid (Y2H) screens when different yeast strains are used (Koegl and Uetz, 2007) (Table I, last row, last column; Figure 2; Supplementary Table S3). Interestingly, when the latter 'consensus' interactions were retested using mating by re-transforming the respective TFs into the mating strain, we obtained a lower validation percentage for mating (70% or 9/13) versus direct transformation (92% or 12/13) (Supplementary Figure S8; Supplementary

Table S3, last row, columns three and six; Materials and methods). This result is consistent with our earlier hypothesis that re-transformation of the TFs into the mating strain lowers the reproducibility, and suggests that re-performing a mating screen with a second and independent yeast TF array may be a more stringent validation strategy. *Third*, we found that 6 out of 15 transformation-specific and 6 out of 12 mating-specific, reproducibly detected interactions were also found when tested on an individual (i.e., not large-scale) basis with the complementary Y1H approach (i.e., mating when originally

found using transformation and *vice versa*). These results indicate that, as expected, both large-scale screening procedures are non-exhaustive (Supplementary Figure S8; Supplementary Table S3, last row, columns two and five). *Fourth*, when only considering reproducibly detected known interactions, we obtained a sensitivity of 18% (5 out of 28) and 25% (7 out of 28) for the transformation and mating procedures, respectively (Table I, last row, columns four and eight). These results are in line with previously reported numbers in Y1H and Y2H screens (Deplancke *et al*, 2006a; Braun *et al*, 2009; Hens *et al*, 2011; Reece-Hoyes *et al*, 2011a). The reason for this small discrepancy in sensitivities between the transformation and mating strategies is unclear and simply may indicate that some of the interactions are more robustly detected in a diploid background.

In general though, we can conclude that the two Y1H screening procedures have a comparable detection performance and feature a similar throughput, as they both could theoretically accommodate several dozen DNA bait screens per month. However, mating may nevertheless be the method of choice for groups wishing to replicate the presented set-ups. This is because, even though it requires more hands-on work compared with automated direct transformation (haploid Y1H), it does not require an elaborate robotic set-up and can be more easily implemented in a laboratory setting.

## Validation of Y1H-detected interactions using motif scanning and luciferase reporter assays

To independently validate the detected interactions and to assess the potential technical precision rate and sensitivity of our Y1H assays, we first compared and contrasted our Y1H results with those derived from motif scanning. For consistency, we used a recently published high-throughput SELEX data set by Jolma *et al* (2013) because it contains the largest collection of high-quality PWMs for human and mouse TFs to date. This allows us to bypass potential issues with the quality of PWMs, which may vary depending on the input data or method through which they were derived (Stormo and Zhao, 2010; Gordan *et al*, 2011; Medina-Rivera *et al*, 2011; Weirauch *et al*, 2013). We were able to retrieve reliable PWM data for 326 out of 750 TFs (43%) within our clone collection, based on the fact that these TFs were directly probed or at least their one-to-one human orthologs (Materials and methods). We then used the widely applied method FIMO (Grant *et al*, 2011), which is part of the MEME software package, to scan the seven bait sequences for the presence of motifs at three different detection stringencies including default parameters (Materials and methods).

Our analyses revealed that using default scanning parameters ($P < 1e$-4), ~50% of reproducibly detected Y1H interactions are supported by respective motif hits (see the 'all' category and Y1H 'positive predictive value (PPV)' column in Supplementary Table S4). This PPV, or the precision rate (Materials and methods), increases to ~90% when motifs are called at lower stringency ($P < 1e$-3), thus providing *in silico* support for the majority of reported interactions. We did not detect any clear emerging pattern regarding which Y1H procedure is most robust, even though consensus interactions

tend to have the highest PPV value at high motif calling thresholds. This indicates that interactions that are driven by binding sites that closely resemble the consensus sequence will likely be detected by both Y1H approaches. Interestingly, the PPV values for our positive control interactions (i.e., those reported in the literature) were only slightly higher than the Y1H ones (Supplementary Table S4), indicating that reproducibly detected Y1H- and literature-based interactions feature a comparable motif occurrence probability. In addition, 16 out of 40 detected Y1H interactions were not predicted through motif scanning. These 16 interactions are mediated by 10 TFs. For 2 out of these 10 TFs, PWMs are available within the Jolma *et al* (2013) data set, yet motifs were not predicted within the respective DNA baits (the *Fos* promoter and the *Mlcrhs4* element for the TF BARX1 and the *Mcts2-Id1* enhancer for the TF ONECUT2). A further scan of other databases (TRANSFAC, Wingender, 2008; JASPAR, Bryne *et al*, 2008; and UNIPROBE, Robasky and Bulyk, 2011) revealed that PWMs appear to be available for five out the remaining eight TFs. This leaves three TFs (CDX4, NR2C1, and ZDHHC9) and four interactions that would not have been detected through regular motif scanning, next to the three interactions involving two TFs for which the corresponding motifs could not be detected within the target sequences despite the availability of the respective PWMs (see also below).

However, many more motifs were called for which the respective TF–DNA interactions were not detected using Y1H. This is revealed when calculating the sensitivity, which represents the percentage of motif-predicted TF–DNA interactions supported by Y1H. We found that the sensitivity amounts to ~2–10% of all reproducibly detected interactions for, respectively, low to high motif calling stringencies (the 'all' category in Supplementary Table S4). These numbers reflect on the one hand the fact that motif scanning may itself be prone to false positive calls of which the rate depends on the type of algorithm, the quality of TF PWMs, or the scanning parameters (e.g., detection threshold) used (Gordan *et al*, 2011; Medina-Rivera *et al*, 2011; Klepper and Drablos, 2013; Weirauch *et al*, 2013), but on the other hand the likely substantial technical false negative rate of the Y1H assay. One possible reason for the low sensitivity of the Y1H assay may be the occlusion of binding sites by nucleosomes. In other words, since the DNA baits are chromatinized, genuine binding sites may not be accessible because of nucleosomal interference.

To test this, we predicted the nucleosome occupancy of each DNA bait using the Kaplan *et al* (2009) model. All seven DNA baits showed an overall high nucleosome occupancy landscape (Supplementary Figure S9; Materials and methods). This suggests that many interactions which involve sites that are predicted to be strongly occupied by nucleosomes are nevertheless detected by Y1H. An example is the interaction between the *Mcts2-Id1* enhancer and RFX2 whose motif is predicted to be located in a high nucleosome occupancy region ($P = 0.88$). This observation is consistent with the PPV values, which gradually decrease upon progressively excluding regions within DNA baits that feature high ($P > 0.7$), moderate to high ($P > 0.5$), or low to high ($P > 0.3$) predicted nucleosome occupancy probabilities (Supplementary Table S5). In other words, by removing regions within DNA baits with a moderate to high predicted nucleosome occupancy, we remove many

motifs that support detected Y1H interactions. However, we observed an inverse relationship between sensitivity values when we considered all Y1H interactions supported by high stringency motifs. This suggests that sites that are predicted to be weakly occupied by nucleosomes and have a high motif score are more likely to be detected with Y1H. Provided that the predicted data reflect true *in vivo* behavior, our analyses point to nucleosome occupancy as another plausible factor affecting the predictive value of our Y1H platform.

We further explored the molecular basis for the probable low sensitivity of the Y1H assay, by testing whether the overall motif load for a specific TF within a DNA element affects the interaction detection probability. We found that the sensitivity of the Y1H assay is indeed partially dependent on both motif number and score in that the higher both are, the more likely the respective interaction will be detected (Supplementary Table S6). We conclude that the number of TF binding sites, their score, and the nucleosome occupancy landscape are all factors that influence the predictive value of the Y1H system.

To additionally validate the detected interactions and to experimentally benchmark our Y1H assays, we also performed systematic TF overexpression-based luciferase reporter assays with all seven screened DNA elements in HEK293 cells. Of the 59 tested interactions including 26 known interactors and 33 novel Y1H-detected interactions, 40 (68%) significantly ($P < 0.05$, $n = 6$; Materials and methods) altered luciferase activity (Figure 3A and B; Supplementary Figure S10; Supplementary Table S3). Among the 40 luciferase-validated interactions, 13 (out of 15 tested TFs, 87%) and 6 (out of 12 tested TFs, 50%) were found by only transformation or mating, respectively, and 10 (out of 13 tested TFs, 77%) by

both approaches. Furthermore, out of 33 novel Y1H-detected interactions, 26 (79%) were positive in HEK293 cells. Of the 19 known interactors that were not detected by Y1H, 11 (58%) were positive (Figure 3C; Supplementary Table S3). These results indicate that interactions detected by either one or both Y1H screening strategies have a similar, if not greater, likelihood of altering luciferase activity than those detected by other methods, validating both presented Y1H approaches.

## MITOMI-based analysis of regulatory elements

Yeast one-hybrid, motif scanning, and luciferase reporter assays do not yield conclusive evidence that the tested TFs are directly binding to their respective DNA elements. In addition, while Y1H assays identify candidate interactors for DNA elements, they do not provide positional information as to where exactly these TFs bind within these relatively long (up to 1 kb) DNA sequences. We therefore elaborated on our recently developed microfluidics-based protein–DNA interaction mapping approach, termed MARE, enabling the fine-grained localization of TFs of interest within specific regulatory elements (Hens *et al*, 2011). The MARE technique can be compared with a series of electrophoretic mobility shift assays (EMSAs), in which a TF is tested for its ability to bind to a collection of typically small DNA sequences, and relative DNA occupancy data for each sequence can be derived (Figure 4A; Materials and methods). Similarly to EMSA, the MARE protocol starts with small DNA elements, resulting from the fragmentation of long regulatory DNA sequences, which are tested individually for binding to a specific set of TFs

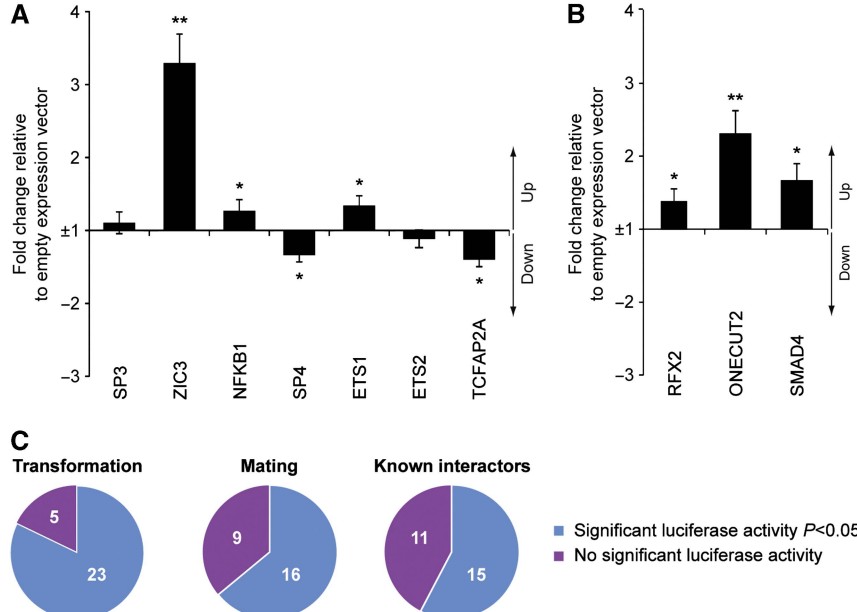

**Figure 3** Luciferase validation of Y1H interactions. Luciferase reporter-based validation with either the *Mmp9* promoter (**A**) or the *Mcts2-Id1* enhancer element (**B**), respectively. The results for the other DNA baits can be found in Supplementary Figure S10. The fold change of the normalized firefly to Renilla ratio compared to the normalized firefly to Renilla ratio of the negative control (empty expression vector) is plotted. Consensus interactions were analyzed as well as those reproducibly found by either mating or transformation. Interactors reported in the literature (Supplementary Table S2) and not found in the Y1H screens were also tested. The error bars represent the standard error of six independent biological experiments. **$P < 0.01$ and $0.01 < *P < 0.05$ compared with the negative control. (**C**) Proportion of reproducibly detected transformation-, or mating-based interactions that scored positive in the luciferase reporter assay. As a comparison, the proportion of known interactions (i.e., positive controls) that altered luciferase activity is also plotted. Source data for this figure is available on the online supplementary information page.

(e.g., those that were identified using our Y1H screens). However, MARE accommodates >700 EMSA-like assays at once on one microfluidic chip in a relatively straightforward and cost-effective manner. This in turn enables the generation of a relative DNA occupancy landscape for each TF of interest over the length of the respective regulatory element where the regions of highest occupancy likely contain the respective TF binding sites (Figure 4B–D). Thus, MARE allows the simultaneous validation and localization of protein–DNA interactions within regulatory elements.

To apply MARE for mouse protein–DNA interaction studies, we tested several DNA bait- and protein expression formats, resulting in the generation of a MARE-compatible vector that enables the wheat germ (WG)-based *in vitro* expression of GFP-tagged proteins (Supplementary Figure S11; Materials and methods). Compared with our initial strategy (Hens *et al*, 2011), we also incorporated a multiplexer into our microfluidics design to enable the simultaneous testing of multiple off-chip produced TFs within the same MARE assay (Materials and methods). In this new strategy, each tagged TF is first *in vitro* expressed in a microcentrifuge tube (off-chip) and then transferred in each specific chip row to test for binding to the spotted DNA sequences (Figure 4A). The advantage of off-chip synthesis is that it yields a larger protein amount relative to on-chip protein expression, which increases the overall signal-to-noise ratio and improves binding site resolution (Supplementary Figure S12). We first benchmarked MARE by analyzing the interaction between NFKB1 and the *Mmp9* promoter. This is because there is solid evidence for one specific NFKB1 binding site within the *Mmp9* promoter based on several assays including ChIP and PWM data (Supplementary Table S2; Materials and methods). For this purpose, we tiled the 998 bp-long *Mmp9* promoter in fragments of 36 bp, each overlapping 24 bp, allowing us to examine the binding of NFKB1 to this element with a maximum resolution of 12 bp. As shown in Figure 4B and Supplementary Figure S13, the MARE-based DNA occupancy landscape revealed two reproducibly detected and significant binding peaks. One is exactly positioned within the NFKB1-bound region identified by previously published methods, while the second contains another predicted strong NFKB1 binding site. These results validated the MARE technique and prompted us to analyze the interactions involving the six other remaining known or Y1H-detected interactors with the *Mmp9* promoter, that is, ETS1, ETS2, TCFAP2A, ZIC3, SP3, and SP4 (Supplementary Figure S13; Supplementary Table 2). Off-chip expression of TCFAP2A was overall low and therefore not tested further. ZIC3 exhibited strong site-specific binding as indicated by the presence of one clear, motif-containing DNA occupancy peak. For both ETS1 and ETS2, we identified three consistent binding peaks by MARE, with the largest being shared between the proteins and overlapping the previously reported binding region (Figure 4C). However, similar to NFKB1, we found that even the smaller and not necessarily significant ETS1 and ETS2 MARE peaks tended to contain predicted motifs. To more formally assess the molecular relevance of this overlap, we plotted the MARE versus the motif scores for each TF. This revealed that the MARE scores follow the motif score landscape well (Supplementary Figures S14 and S15). In fact, they do so for the majority of tested TFs,

although the correlation tends to deteriorate for very low scoring motifs, many of which may therefore constitute false positive calls. This is particularly well illustrated by the SP3 and SP4 MARE data (Supplementary Figure S14). Plotting these data revealed more distributed (e.g., compared with NFKB1), but largely overlapping DNA binding profiles (Figure 5A; Supplementary Figure S13), whereby most peaks contain predicted SP3 and SP4 DNA binding motifs. This overlap is consistent with the highly conserved nature of the DNA binding domains (DBDs) of all SP proteins (i.e., SP1, SP3, and SP4) (Figure 5B) and with the observation that these proteins can compete for the same DNA sites (Kaczynski *et al*, 2003). In sum, both TFs bind DNA in a correlated and site-specific manner with a preference for SP-like motifs and this DNA binding behavior is accurately captured by the MARE assay.

Finally, we used MARE to validate and characterize Y1H interactions detected with the orphan enhancer *Mcts2-Id1*. Both RFX2 and SMAD4 showed site-specific binding in MARE, again often overlapping with predicted binding sites (Figure 4D; Supplementary Figure S16). The interaction involving ONECUT2 is of special interest since it was not predicted using motif scanning despite the availability of a ONECUT2 PWM. However, we observed two reproducibly detected peaks, providing additional support for the Y1H data. We have listed the sequences through which ONECUT2 likely interacts with the *Mcts2-Id1* enhancer in Supplementary Table S7, which also features the sequences for interactions involving other TFs (ETS1 and ETS2), which do not overlap with predicted motifs. A more detailed dissection of these sequences will now be required to identify the precise molecular mechanisms and binding sites underlying the observed TF–DNA interactions. For example, we observed that a reproducibly detected ETS1 peak is not supported by a motif call (Figure 4C; Supplementary Figure S13). Upon closer inspection, we found that an ETS-like binding site is present, but its probability ($P = 1.2e\text{-}3$) fell just below our lenient detection threshold of $P < 1e\text{-}3$. Intriguingly, when the underlying PWM was replaced by a model from TRANSFAC, the motif was scored as significant ($P = 2.2e\text{-}4$), illustrating the difficulty in obtaining an accurate snapshot of the DNA binding landscape based on motif calling alone.

In conclusion, we found that four out of four testable, known interactors (i.e., NFKB1, ETS1, ETS2, and SP3) and nine out of nine Y1H-detected interactions showed at least one site-specific, reproducibly detected binding peak, providing strong support for the observed Y1H interactions as well as the utility of the MARE assay to resolve DNA binding patterns for specific TFs with DNA elements of interest.

### *In vivo* validation of detected TF–enhancer interactions

The approaches utilized above do not provide information about whether the detected TF–DNA interactions take place *in vivo*. Consequently, it would be optimal to obtain additional data supporting their occurrence in a biologically relevant context. *First*, given the remarkable DNA binding patterns observed for SP3 and SP4 in the MARE assay (Figure 5A), we

aimed to validate the binding of these TFs to the *Mmp9* promoter by performing ChIP in NIH-3T3 cells using NFKB1 as a positive control. Specifically, it was shown that NFKB1

interacts with the *Mmp9* promoter in NIH-3T3 cells after TNF-α induction (Balasubramanian *et al*, 2011), and we therefore tested for binding of SP3, SP4, and NFKB1 to *Mmp9* by

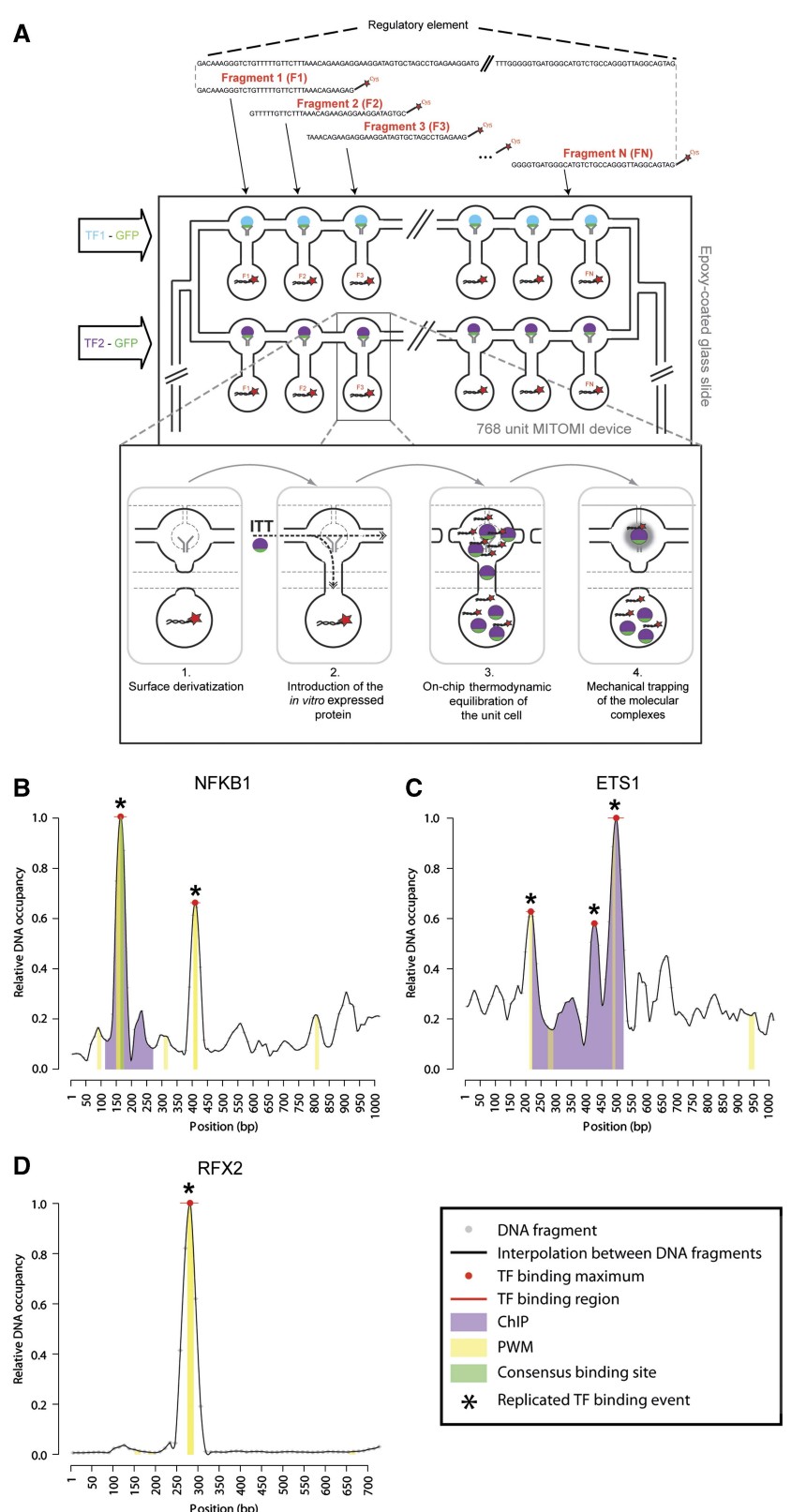

ChIP-qPCR in TNF-α-stimulated and unstimulated cells. We confirmed binding of all three TFs to the *Mmp9* promoter after TNF-α stimulation, whereas little to no binding could be detected in unstimulated cells (Supplementary Figure S17). The occurrence of multiple SP-like motifs within the *Mmp9* promoter and the apparent recruitment of SP3/SP4 to those homotypic binding sites is intriguing and implies either additive/synergistic effects in the regulation of *Mmp9* gene expression or functional redundancy against deleterious mutations (Maurano *et al*, 2012; Spivakov *et al*, 2012). Interestingly, Pascal and Tjian (1991) revealed in a classic study that adjacent SP1 binding sites promote the formation of higher order SP1 complexes. These complexes are able to interact more efficiently with the general transcriptional machinery than individual SP1 proteins resulting in higher transcription, thus lending support for the synergistic nature of the detected SP3/4 binding events with the *Mmp9* promoter.

In a second experiment, we decided to take advantage of the information provided by the MARE assay to *in vivo* validate

protein–DNA interactions in a DNA-centric manner. For this purpose, we focused on the *Mcts2-Id1* enhancer as its activity has been assessed in mouse embryos in the VISTA Enhancer resource (Visel *et al*, 2007) at embryonic day (E) 11.5, revealing expression in the eye, the forebrain, the midbrain, and the hindbrain. Specifically, we set out to assess whether a small region (∼50 bp) within this enhancer identified by MARE analysis to contain or overlap with DNA occupancy peaks for RFX2 and ONECUT2 is necessary for the observed expression pattern of the enhancer (Figure 4D; Supplementary Figure S16). This is because, even though this DNA segment exhibits poorer sequence conservation compared with other segments within this enhancer (Supplementary Figure S18), we made several independent observations that all point to the importance of this region in regulating the activity of the *Mcts2-Id1* enhancer through the RFX2 and ONECUT2 TFs.

First, we performed Y1H assays with the enhancer minus the 50 bp region (enhancer Δ50) and found that RFX2 and ONECUT2 no longer bind to this deletion compared with the

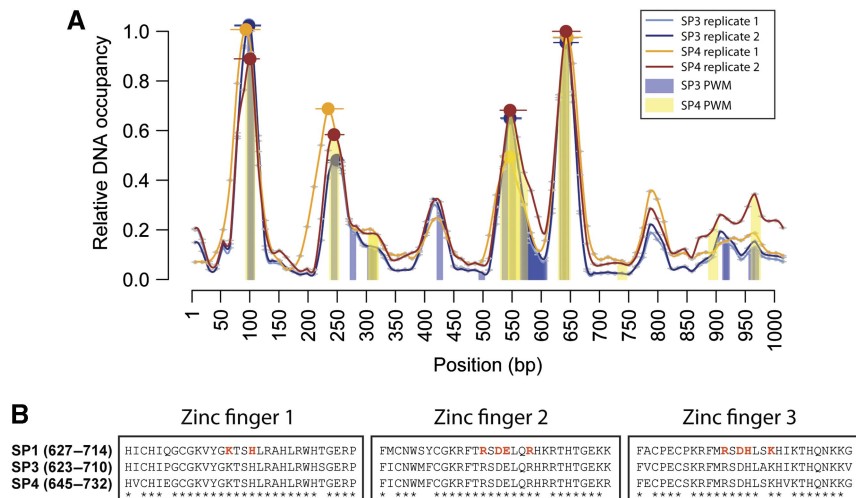

**Figure 5** Shared DNA occupancy landscape between SP3 and SP4 across the *Mmp9* promoter element. (**A**) MARE-based analysis of Y1H-detected protein–DNA interactions involving the TFs SP3 and SP4 and the *Mmp9* promoter. For each replicate, the relative DNA occupancy values are plotted as described in Figure 4. SP3 replicate 1, SP3 replicate 2, SP4 replicate 1, and SP4 replicate 2 are represented by light blue, dark blue, orange, and brown lines, respectively. The PWM-based binding site predictions for SP3 and SP4 proteins are indicated with blue and yellow bars. (**B**) Multiple protein alignment of the annotated SP1 DNA binding domain with SP3 and SP4. Star symbols indicate identical amino acids across all three proteins. Amino acids highlighted in red indicate the proposed protein–DNA interaction residues of the human SP1 protein (Oka *et al*, 2004). The SP1 DNA binding domain between human and mouse is highly conserved with only a single amino-acid change between both proteins.

**Figure 4** MARE pipeline and results. (**A**) Schematic picture of the MARE principle. Cy5-labeled, double-stranded DNA fragments derived from the regulatory element of interest are spotted on an epoxy glass slide. The glass slide is then aligned with a MITOMI chip such that each microfluidic unit cell contains only one fragment of the regulatory element. The expression templates coding for TFs were incubated off-chip with an ITT mix after which the resulting proteins were introduced in different rows of the MITOMI chip using a multiplexer design. The dynamic protein–DNA interaction detection steps are described in the inlet. (1) The surface chemistry is derivatized as described in Maerkl and Quake (2009) allowing the immobilization of biotinylated-GFP antibody at the center of the upper unit. (2) The *in vitro* translated protein is introduced into the units. (3) The microfluidic chip is then incubated for 1 h to allow the thermodynamic equilibration of putative TF–DNA interactions. (4) TF–DNA interactions are pulled down in the center region of the upper unit by the biotinylated GFP-antibody and then mechanically trapped by closing a microfluidically controlled 'button' membrane, which removes all surrounding solution phase molecules after washing. The TF (GFP) and DNA element (Cy5) fluorescence is then quantified with a microarray-like fluorescent scanner, enabling the assessment of how much TF and DNA element is trapped. Note that here, only one protein binding to one DNA fragment is visualized. This is a simplification as in reality many TFs are pulled down in each unit, and all of them are able to bind to the respective DNA fragments. (**B–D**) Examples of MARE-based protein–DNA interaction analyses with individual fragments derived from the *Mmp9* promoter element involving NFKB1 (B) and ETS1 (C) or with fragments derived from the *Mcts2-Id1* enhancer element involving RFX2 (D). Data for the other tested interactions are shown in Supplementary Figures S13 and S16. Bound DNA levels normalized over surface-immobilized protein amounts are plotted for each 12 bp nucleotide stretch as small gray dots with horizontal lines indicating the 12 bp region. Signals between every 12 bp nucleotide were determined by interpolation. Significant sequence-specific binding region peaks are pointed out with a red line, while binding region maxima are highlighted with red dots. Peaks found in two independent experiments are marked with an asterisk. When applicable, a ChIP-based region (ETS1 and ETS2; Wei *et al*, 2009) and position weight matrix (PWM)-based binding site predictions are indicated with purple and yellow bars, respectively. The consensus NFKB1 binding region is indicated by a green bar.

wild-type construct, in contrast to SMAD4, which binds elsewhere in the enhancer as derived from our MARE analysis (Figures 4D and 6A; Supplementary Figure S16). These results clearly support the observation that this 50 bp region mediates the binding of both RFX2 and ONECUT2 to the *Mcts2-Id1* enhancer. Second, luciferase reporter assays revealed a significant decrease in luciferase activity with the deletion compared with the wild-type construct upon RFX2 or ONE-CUT2 overexpression, again in contrast to SMAD4, providing additional support for our observations (Figure 6B). Finally, the involvement of RFX2 and ONECUT2 in controlling the activity of the *Mcts2-Id1* enhancer is intuitive as they are both expressed between E10 and 11.5 in tissues in which the enhancer also exhibits activity (EMAGE gene expression database, Richardson *et al*, 2010; the Mouse Genome Database (MGI 5.11, May 2012), Finger *et al*, 2011). For example, *Rfx2* has been shown to be expressed in the forebrain, eye, and the heart (among other tissues), whereas *Onecut2* is also expressed in the developing brain and the gut next to the embryonic spinal cord and the dorsal root ganglion (Richardson *et al*, 2010; Finger *et al*, 2011) (Figure 6C).

Given the multi-tiered evidence pointing to the role of this 50 bp region in mediating *Mcts2-Id1* enhancer activity, we decided to test the *in vivo* impact of its deletion. First, we analyzed the wild-type enhancer expression pattern via lentivector-mediated transgenesis at E10.5 (Materials and methods; Friedli *et al*, 2010), given the fact that both RFX2 and ONECUT2 were shown to be highly expressed at this time point. Nine out of eleven transgenic embryos exhibited a similar expression pattern with *lacZ* staining in the forebrain, the nasal placode (part of the forebrain), the eye, the heart, the dorsal root ganglion, and the limb which together closely resembled the previously described expression pattern at E11.5. We also observed expression in the embryonic gut, which was not previously reported (Figure 6C; Supplementary Figure S19). Interestingly, deletion of only 50 bp substantially altered this pattern as, for five out of eight embryos, we no longer observed activity in the forebrain, the eye, the nasal placode, the heart, and the future gut, regions in which at least one of the two TFs of interest is also expressed (Figure 6C).

Together, these results support a model in which RFX2 and ONECUT2 cooperate to regulate the activity of the *Mcts2-Id1*

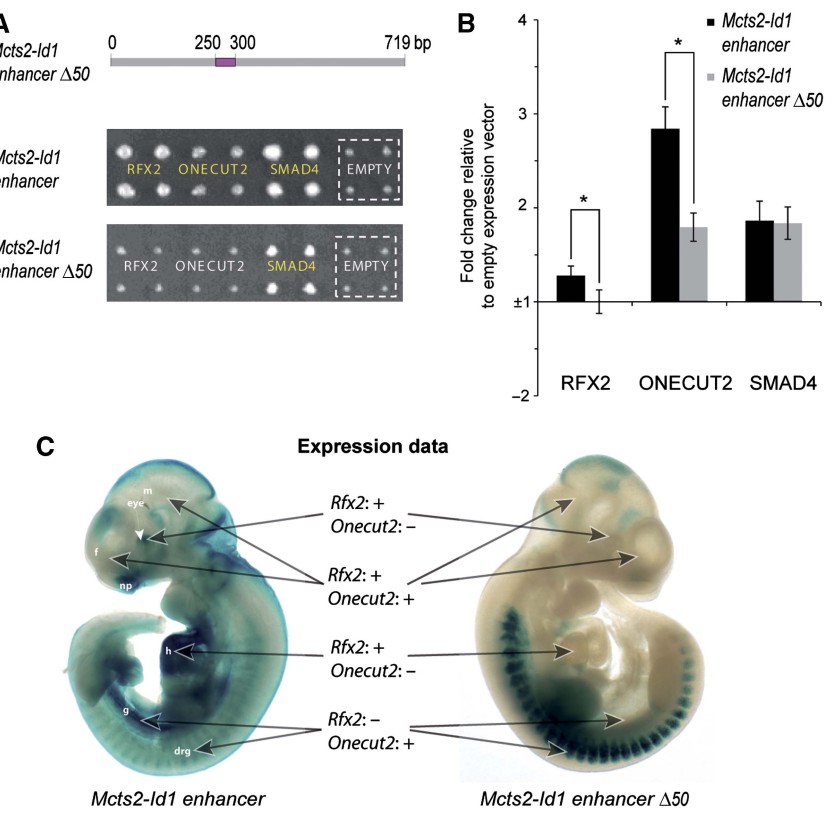

**Figure 6** RFX2 and ONECUT2 regulate the activity of the *Mcts2-Id1* enhancer through a 50 bp region. (**A**). Manual transformation experiments of RFX2 and ONECUT2 with either the wild type or the 50 bp deletion (Δ50, highlighted in purple) *Mcts2-Id1* enhancer element. Yeast strains were spotted in quadruplet. Positive interactions are highlighted in yellow and can be compared with a negative control (original pAD-DEST vector, termed 'empty' and indicated by a dashed square) illustrating the background growth for each DNA bait. The baits were selected on 10 mM 3-AT. (**B**) Luciferase reporter-based validation with either the wild type or the Δ50 enhancer element. HEK293 cells were transiently co-transfected with DNA bait reporter construct, each of the respective TFs, and the Renilla luciferase vector. The fold change of the normalized firefly to Renilla ratio compared with the normalized firefly to Renilla ratio of the negative control (empty expression vector) is plotted. The error bars represent the standard error of 10 independent biological experiments. $0.01 < *P < 0.05$ compared with the wild-type construct. (**C**) LacZ staining results of E10.5 transgenic embryos containing either the wild type (left) or the Δ50 (right) *Mcts2-Id1* enhancer-lacZ reporter construct. The expression profiles of RFX2 and ONECUT2 in the highlighted organs as derived from the literature are summarized by ' + ' (expression) or ' − ' (no evidence of expression). These two embryos contained two integrated lentiviral DNA copies each. f, forebrain; m, midbrain; np, nasal placode; h, heart; g, future midgut; drg, dorsal root ganglion. Source data for this figure is available on the online supplementary information page.

enhancer in several embryonic tissues through a small 50 bp region with their specific, regulatory contributions depending on the respective tissue.

## Discussion

In this study, we established a comprehensive mouse TF ORF clone collection in a versatile Gateway format, which we are still actively expanding through gene synthesis of missing TF ORFs. We anticipate that this resource will therefore soon be similar in size to its human counterpart (Hu *et al*, 2009) and feature a similar utility in enabling the implementation of different techniques or experiments to study mammalian gene regulatory networks.

Here, we used this resource to develop a cross-platform pipeline to first detect and then characterize mammalian protein–DNA interactions at relatively high throughput and resolution (Figure 7). We believe that this pipeline has several advantages over *in silico* protein–DNA interaction detection approaches: *first*, our pipeline allows the screening of virtually every TF while motif scanning is limited to those TFs for which the DNA binding specificities (PWMs) have been characterized. For example, 10% of our detected Y1H

interactions involved TFs for which to our knowledge no PWM is available. *Second*, motif scanning is more complex than may seem at first sight. For example, the quality of PWMs is variable and several PWMs may be associated with the same TF (Stormo and Zhao, 2010; Gordan *et al*, 2011; Medina-Rivera *et al*, 2011; Weirauch *et al*, 2013), as also shown in our analyses, making it difficult to compile the most accurate or comprehensive data set. Perhaps even more importantly, about half of the detected Y1H interactions would have been missed with default motif scanning parameters. One option would be to lower the required motif score because almost all detected Y1H interactions were supported by motifs detected with a lower stringency and because the MARE and motif scores correlate well. However, lowering the detection threshold increases the motif overcalling risk, resulting in a higher number of false positives. We found that the sensitivity of the Y1H assay is ~3% at a default motif calling stringency. This means that ~60 reactions are predicted per DNA bait when the latter are scanned for motifs using a default threshold. While the Y1H assay has likely a substantial false negative rate (see also below), it is unlikely that all of these predicted interactions constitute true positives due to ambiguity in TF PWMs, motif scanning algorithms, motif

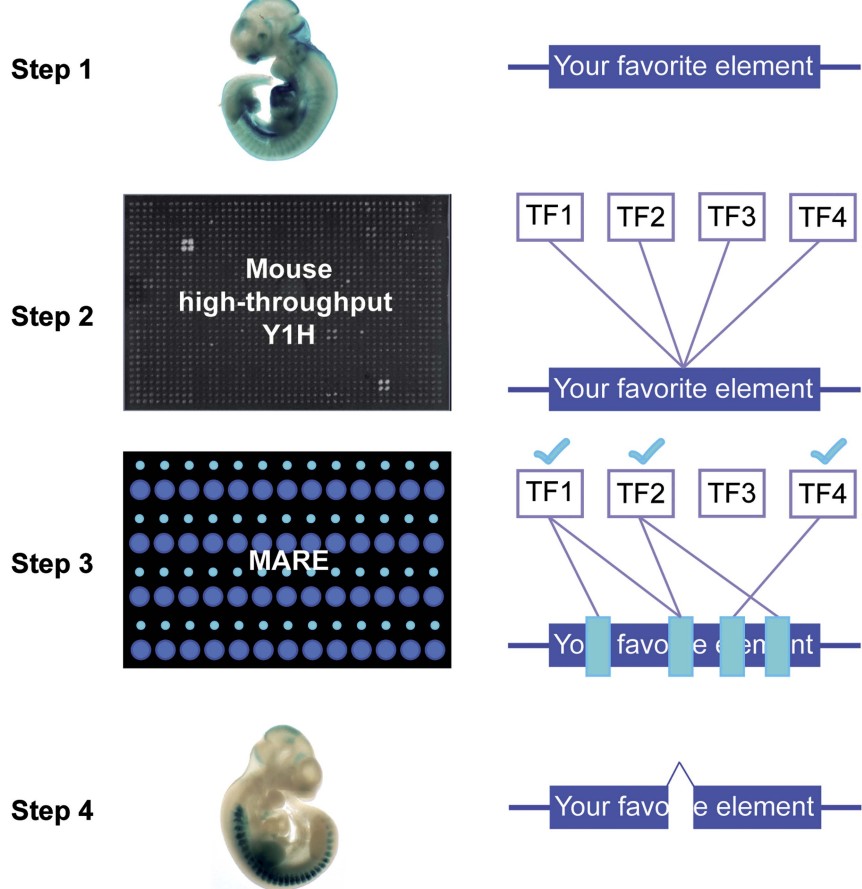

**Figure 7** Schematic overview of the pipeline employed to deorphanize mammalian gene regulatory elements. The regulatory element of interest is first cloned (Step 1), and then integrated into yeast to enable high-throughput Y1H screens leading to the identification of putatively interacting TFs (Step 2). In Step 3, MARE analysis is performed to both validate (reflected by light blue check marks) and map the detected TF–DNA interactions (indicated by light blue boxes) within the respective regulatory element. Finally, small binding regions of interest can be deleted to examine the relevance of these DNA segments in mediating the *in vivo* activity of the regulatory element (Step 4).

scanning detection thresholds, and background models. *Third*, the points discussed above highlight the need for experimentally validated motif predictions, which the Y1H and MARE assays readily provide. A reasonable strategy therefore may be to combine motif calling with Y1H screens to yield the most comprehensive and robust set of protein–DNA interactions. These can then be further characterized using the MARE assay, highlighting the unique and complementary nature of the presented pipeline compared with motif calling alone.

We made several important observations regarding the detection performance of the Y1H assay. Specifically, we found that it has a low sensitivity (i.e., 18–25%, based on interactions reported in the literature; and 2–10%, based on motif scanning), but a high precision (~90%, based on motif scanning). These observations suggest that the main weakness of the Y1H assay may be its low sensitivity and not necessarily its low precision as has so far been commonly assumed. The low sensitivity is not unexpected as these values fall within the range reported by other yeast-based one-hybrid or two-hybrid studies (Deplancke *et al*, 2006a; Braun *et al*, 2009; Chen *et al*, 2010; Hens *et al*, 2011; Reece-Hoyes *et al*, 2011a). Indeed, there are several reasons why the Y1H assay may fail to detect specific protein–DNA interactions. For example, we showed that the nucleosome occupancy landscape, the number of TF binding sites per DNA bait, and score of motifs within DNA baits affect the detection probability. Other causes for the low sensitivity may include the inability to detect binding of TFs that require an interaction partner (either another TF or a co-regulator) to bind to DNA, no or low expression of certain mouse TFs in the DNA bait-containing yeast cells, and the fact that some of the interactions may involve a specific TF isoform, which is not included in our TF ORF library. In addition, the post-translational modification (PTM) state of a protein expressed in the yeast cell compared with the endogenous condition may be different. Several types of PTMs such as phosphorylation (e.g., Rivera *et al*, 1993; Cowley and Graves, 2000; Lee *et al*, 2010), sumoylation (e.g., Chou *et al*, 2007; Wei *et al*, 2007; Yan *et al*, 2010), and acetylation (e.g., Hayakawa *et al*, 2004; Sun *et al*, 2009; Waby *et al*, 2010) have been shown to influence TF binding. In this regard, it is interesting that PTMs of at least two TFs involved in three positive control interactions that were not found by the Y1H screen have been shown to influence DNA binding of these TFs (i.e., acetylation of GATA1, Lamonica *et al*, 2006; sumoylation of POU5F1, Wei *et al*, 2007). Nevertheless, our estimation of the sensitivity is also highly influenced by the quality of the reported interactions, which we consider as positive controls. Indeed, in contrast to protein–protein interaction detection assays (Braun *et al*, 2009), there is currently no gold standard collection to evaluate the efficacy of protein–DNA interaction detection methods. It is therefore conceivable that some of the interactions that were not recovered by Y1H, and that we count as false negatives, are in fact false positives from another study. Our observation that interactions retrieved from the literature do not show a greater validation frequency in luciferase reporter assays and only have slightly greater precision rates compared with all reproducibly detected Y1H interactions is consistent with this view. We therefore

believe that our reported sensitivities are conservative estimates.

The precision of Y1H assays had so far not been systematically assessed, but has typically been predicted to be low. Here, we provide evidence that, at least from a technical point of view, this rate is likely high. *First*, we found that the majority of detected interactions involve TFs whose corresponding binding sites are present in their respective DNA baits. *Second*, we tested the interactions using luciferase-based cell-culture assays. While we cannot exclude that some of the observed reporter expression changes were due to indirect effects, our results nonetheless suggest that the majority of tested protein–DNA pairs can also interact in mammalian cells. *Third*, we evaluated and characterized Y1H-detected interactions using the microfluidics-based MARE assay, which we show tracks the DNA binding behavior of TFs well. Similarly to the luciferase-based assay, we were again able to validate most interactions based on the fact that we observed sequence-specific binding of identified TFs with the respective DNA elements. These results indicate that the majority of Y1H-detected interactions can be corroborated by *in silico*, *in vitro*, as well as cell culture-based assays, and that at least from a biochemical point of view, most of the Y1H-detected interactions appear to be true positives.

Whether these interactions also occur *in vivo* is more difficult to systematically assess as interactions may for example take place in only a few cells, or during only a brief time period (Deplancke, 2009). Nevertheless, we provide experimental evidence that at least several novel Y1H-identified interactions have potential *in vivo* relevance, notably those involving the *Mmp9* promoter and the *Mcts2-Id1* enhancer. The latter regulatory element is located between ~0.7 and 1.4 kb upstream of the gene coding for the HLH domain-containing inhibitor of DNA binding 1 protein, *Id1*, and yields a virtually identical expression pattern as *Id1* based on findings by Jen *et al* (1996), suggesting that it mediates its transcription. Here, we exploited the mapping resolution offered by MARE to delineate a small DNA segment within this enhancer, which controls the *in vivo* activity of this regulatory element. Interestingly, conservation analysis would not necessarily have identified this specific DNA segment, since it exhibits poorer conservation than its surrounding sequences. This observation appears consistent with the findings from the ENCODE project which also reported modest base pair overlap between experimentally derived and conservation-based regulatory annotations (Encode Consortium, 2007). Using our Y1H assay, we identified two TFs binding to this small DNA region: (1) RFX2 is an RFX-type winged-helix domain-containing transactivator (Horvath *et al*, 2009), which is still poorly characterized, despite being expressed in a wide range of tissues as indicated above; (2) ONECUT2 is a homeodomain-containing TF, which, similar to RFX2, is expressed in many tissues. However, their expression patterns do not completely overlap, nor with those of the enhancer or *Id1*, suggesting that, while we cannot exclude cooperativity in some tissues, they each may individually contribute to the activity of the enhancer (and by extension the expression of *Id1*) in specific tissues such as the heart (RFX2) and the gut (ONECUT2). The involvement of ONECUT2 in regulating gene expression in the developing gut is supported by recent studies

reporting the delineation of specific antero-posterior regions by this TF, its presence in several gut cell types later in development, and altered intestinal gene expression in *Onecut2* knockout mice (Vanhorenbeeck *et al*, 2007; Dusing *et al*, 2010). Together, our data point to a role for ONECUT2 in controlling gut expression of *Id1*, which itself acts as a proliferative and tumorigenic factor in this tissue (Ruzinova and Benezra, 2003). Another TF binding to the *Mcts2-Id1* enhancer, albeit not to the same region as ONECUT2 and RFX2, is SMAD4. Interestingly, SMAD4 has already been shown to control the expression of the human *Id1* gene, so it appears reasonable to assume that this regulatory interaction also occurs in the mouse (Katagiri *et al*, 2002). Thus, the data presented in this study strongly support the *in vivo* relevance of all three detected interactions involving RFX2, ONECUT2, and SMAD4 with the mouse *Mcts2-Id1* enhancer. In addition, they reveal how a small, poorly conserved DNA segment within the *Mcts2-Id1* element is responsible for expression activity in multiple tissues through distinct TF configurations, providing interesting insights into the regulatory architecture of enhancers.

We also found evidence supporting the validity of other detected interactions. For example, we observed binding of the closely related forkhead TFs FOXJ2 and FOXJ3 to the DNase I-hypersensitive site HS 4 of the locus control region (LCR) of the β-globin locus. This LCR not only regulates the expression of β-globin, but also of ε-, Gγ-, Aγ-, and δ-globin. Interestingly, siRNA-mediated knockdown of human FOXJ2 in K562 cells decreased expression of γ-globin mRNA in this cell line (Yang *et al*, 2009), consistent with our observation that this TF upregulates LCR HS4-mediated luciferase expression. These data therefore suggest that the interactions between FOXJ2 and possibly FOXJ3 and β-globin LCR HS4 as observed in our Y1H assay have *in vivo* relevance. Finally, we also detected binding of the homeobox TF CDX4 to the same DNase I-hypersensitive site of the control region of the β-globin locus. As there are numerous reports for the role of CDX4 in hematopoiesis (e.g., Wang *et al*, 2005; Bansal *et al*, 2006; Lengerke *et al*, 2007), it is plausible that this TF directly regulates the expression of one or more globin genes, given their importance in red blood cell function.

In conclusion, we believe that the presented two-tiered approach provides a valuable strategy to identify and characterize protein–DNA interactions with regulatory elements of interest that may be relevant *in vivo*. The resulting data can then be used to generate and model gene regulatory networks, which should be of great value to decipher the regulatory grammar underlying mammalian gene expression.

# Materials and methods

## Gateway cloning of mouse TF ORFs

The first step toward the generation of the mouse TF ORF clone resource was to create a comprehensive list of predicted mouse TF-coding genes. Using data from the DBD database (Wilson *et al*, 2008) and from a complementary study (Gray *et al*, 2004), which also aimed at identifying all sequence-specific DBD proteins, we predict that the mouse genome contains 1576 TF-coding genes. We were able to retrieve cDNA templates for 1260 of these from various sources, including the FANTOM Consortium (RIKEN, Japan), Source Bioscience (UK), and Open Biosystems (USA). To initiate the cloning of the corresponding ORFs, we first applied a strong selection scheme

choosing the longest ORF per TF containing the respective DBD. Specifically, we confirmed the presence of a DBD by screening the longest ORFs against a subset of the Pfam protein database containing predicted DBDs ($E$-value $< 0.01$). Accepted longest ORFs were then compared with corresponding sequences in the RefSeq database (Release 28). In most instances, the sequences matched perfectly. However, in particular cases, manual curation and sequence adjustment were necessary to ensure correct frame reading as well as sequence completeness. After filtering, 1218 TF cDNA templates were retained. TF ORFs without the stop codon were then PCR amplified as described (Hens *et al*, 2011), using primers containing the attB1 and attB2 gateway tails at the 5' end of the forward and reverse primer, respectively (sequences are listed in Supplementary Table S1). When the PCR amplification using cDNA clones as template was reproducibly not successful, an RT–PCR strategy was adopted to retrieve additional TF ORFs by extracting total RNA from mouse 3T3-L1 cells using the RNeasy mini kit (Qiagen). One microgram of total RNA was used as an input to generate cDNA using the SuperScript III First-Strand Synthesis kit (Invitrogen). Successfully PCR-amplified TF ORFs were cloned into the pDONR221 vector using Gateway cloning by mixing 100 μg of the pDONR221 vector, 2 μl of the PCR product, and 0.5 μl of BP clonase II enzyme mix (Invitrogen). After incubating for 18 h at 25°C, this mix was transformed into competent DH5α cells and single colonies, typically six per TF, were analyzed by colony PCR with M13F and M13R primers using standard protocols. The TFs that were successfully cloned in pDONR221 (further called TF entry clones) were subsequently analyzed by high-throughput sequencing.

## High-throughput sequencing of TF clone ORFs

Library preparation, high-throughput sequencing, and data analysis were performed as described (Hens *et al*, 2011). Briefly, TF entry clones were pooled equimolarly and fragmented. Sequencing libraries were prepared with these fragments and sequenced using the Illumina Genome Analyzer II DNA Sequencing Platform (Illumina). The output data were processed using the Genome Analyzer Pipeline Software v1.4 and the resulting file containing the short reads was submitted to the WebPrInSeS server (Massouras *et al*, 2010) together with a file containing the reference sequences for the assembly of the short reads and evaluation of the resulting ORFs in comparison with the respective reference sequences. After the analysis, the clones were labeled as 'Gold', 'Silver', and 'Bronze' depending on their sequence quality. Clones were rejected if they contained partial ORFs, non-sense mutations, missense mutations in a known functional protein domain or more than five missense mutations in total compared with the reference sequence. 'Gold' clones (715 clones) are fully covered by sequencing, 'Silver' clones (35 clones) have the 5' and 3' ends covered by sequencing (i.e., standard ORFeome quality) and finally 'Bronze' clones were only partially sequenced.

## Shuttling the TF ORFs to Gateway compatible AD vectors

The 750 TF ORFs, labeled as 'Gold' and 'Silver' (labels explained in the high-throughput sequencing section), were subcloned from the TF entry clones into an equimolar mix of pAD-Dest-ARS/CEN and AD-Dest-2μ by mixing 2 μl of the TF entry clone, 100 ng of the pAD-Dest mix and 0.5 μl of LR clonase II enzyme mix (Invitrogen). After incubating for 18 h at 25°C, this mix was transformed into competent DH5α cells. Successfully subcloned TFs (745) were miniprepped using the NucleoSpin 96 Plasmid miniprep kit (Macherey-Nagel), typically yielding a concentration of ~100–160 ng/μl and arrayed into two 384-well plates (Supplementary Table S1). Some of the empty wells in the two 384-well AD-TF plates were filled with the original pAD-DEST vectors as negative controls.

## Cloning of the DNA baits

DNA baits were PCR amplified using primers containing restriction enzyme recognition sites at the 5' end of the forward and reverse

primer, respectively (Supplementary Table S2), cloned in the pENTRY-5′ vector using standard restriction-ligation techniques and transferred by Gateway LR cloning into the pMW2 ('*HIS3*') vector (Deplancke *et al*, 2006b). Genomic integrations of the DNA bait-reporter constructs in the YM4271 yeast strain (Deplancke *et al*, 2004) were performed using lithium acetate (LiAc)–polyethylene glycol (PEG) transformation followed by selection on corresponding selective plates. Note that we opted not to include a second reporter such as *lacZ* but rather to use growth on selective medium and reproducibility as our main criteria to call positives. This is because results from Y1H-based protein–DNA interaction screens in *Drosophila* revealed that the *lacZ* reporter often yielded a set of positives that entirely overlapped with the *HIS3*-based list of consensus interactions (Hens *et al*, 2011). Moreover, we found the *lacZ* reporter to be less sensitive than *HIS3* as it for example failed to detect interactions for which we obtained clear evidence that they are relevant *in vivo*.

## Selection of the positive controls for DNA bait interactions

The DNA baits were selected based on their involvement in at least one TF–DNA interaction for which there is strong experimental evidence based on the existing literature (Supplementary Table S2). For each promoter, positive control interactions were retrieved from the TRED database (last query was performed at the end of June 2011; Jiang *et al*, 2007) by selecting promoters by gene name on the promoter retrieval webpage. Specifically, all papers associated with the respective promoters listed in TRED were browsed and positive interactions were added if supporting experimental data were available for these interactions in the respective papers. Interactions for LCRs (e.g., of the β-globin gene) were obtained by extensively browsing the corresponding literature. Enhancer elements were randomly selected from the VISTA Enhancer Database (Visel *et al*, 2007) with the sole criterion that they exhibited *in vivo* enhancer activity (Supplementary Table S2).

## Automated, large-scale yeast transformation

Yeast transformation was performed as described (Hens *et al*, 2011). Briefly, after transformation, yeast strains were allowed to grow for 3 days at 30°C on permissive plates before the colonies were transferred onto selective plates containing varying 3-amino-1,2,4-triazole (3-AT) concentrations (typically from 5 to 20 mM) using the RoToR HDA robot (Singer Instruments). During this transfer, each colony was quadruplicated in a square pattern, yielding four technical replicates and facilitating detection of positive interactions. The plates were typically analyzed after 5–7 days of incubation at 30°C.

## Mating protocol

Mating experiments were performed as described previously (Walhout *et al*, 2000; Vermeirssen *et al*, 2007). Briefly, the two 384-well AD-TF plates were transformed into the Yα1867 strain (a kind gift from John Reece-Hoyes and Marian Walhout, University of Massachusetts Medical School, USA). The transformed yeasts were spotted on Sc-Trp plates and were allowed to grow for 2 days at 30°C. Each colony was then spotted four times to create a 1536 array of transformed yeast on Sc–Trp plates using a RoToR HDA robot (Singer Instruments). Glycerol stocks were prepared for each transformant, and fresh plates were grown from these glycerol stocks every 2 months to avoid cross-contamination caused by repeated replication of the yeast colonies. Two days before the mating, a lawn of the DNA bait strain was prepared on YPD plates and the two AD-TF plates were freshly replicated on Sc–Trp plates. Mating was performed by transferring the AD-TF colonies onto a YPD plate and printing the DNA bait strain on top. The mating plates were left overnight at 30°C before transferring the mated yeast to plates selecting for yeast containing both the reporter construct and the respective TF after which these plates were incubated for 2 days at 30°C. The resulting diploid yeast colonies were then transferred onto selective plates containing varying 3-AT concentrations (usually 5 and 10 mM). The plates were typically analyzed after 5–7 days of incubation at 30°C.

## Semi-automated detection of positive interactions

To avoid biases associated with the manual calling of putative positive interactions, selective plates were analyzed using the TIDY software developed by Hens *et al* (2011), which enables the semi-automated detection of positives. Briefly, interactions of which the convoluted growth score of the respective TF yeast quadrant scored 20% above the highest background intensity value were retained. In roughly one third of the called interactions, this default threshold was lowered to 10% to include positives that bordered the cluster of called positives and that as such still clearly exhibited superior growth than the highest background levels.

## Transient transfection and luciferase assays

For cloning convenience, a Gateway-compatible pGL3-promoter vector was created by digesting the pGL3-promoter vector (Promega) with *acc65*II and *Bgl*II, and blunting the resulting sticky ends using standard techniques. The Gateway AttR4-AttL1 cassette was dropped from the pMW2 vector using *Spe*I and *Sac*II, blunted, ligated into the linearized pGL3-promoter vector and transformed into DB3.1 *ccdB*-resistant cells. The DNA baits were subcloned from the pENTRY-5′ vector into the Gateway-compatible pGL3-promoter vector using a Gateway LR reaction as described under the section '*Shuttling the TF ORF to Gateway compatible AD vectors*'.

TFs were subcloned by a Gateway LR reaction into a Tet-On Gateway-compatible expression vector (a kind gift from Andrea Corsinotti, EPFL, Switzerland). We did not obtain clones for two positive control TFs, Ahr and RXRα, explaining why only 26 and not 28 positive control interactions were tested. One day before transfection, 2.10e4 HEK293T cells were plated on 96-well flat clear bottom white polystyrene plates (Sigma) coated with poly-D-lysine hydrobromide (Sigma) according to the manufacturer's protocol. Before transfection, the medium was changed to a low antibiotic Dulbecco's modified Eagle's medium (with 0.1% of penicillin/streptomycin) containing doxycycline (1 μg/ml, Sigma). Cells were transiently transfected with the DNA bait- and TF-containing clones, together with a Renilla reporter vector (pRL-SV40, Promega) to normalize for transfection efficiency. The transfection mixture was as follows: 550 ng of DNA bait, 400 ng of the TF-expression vector, 50 ng of Renilla reporter vector, supplemented to 50 μl with OPTI-MEM (Invitrogen) and 2.5 μl of FuGENE HD Transfection Reagent (Promega). Forty-eight hours after transfection, the cells were washed with PBS solution, lysed, and luciferase activity was measured using the Dual-Luciferase Reporter Assay System (Promega) on an Infinite f500 microplate reader (Tecan) according to the manufacturer's protocol. Luciferase activity was normalized using the Renilla luciferase activity measurements for each well. Data were calculated as fold change in RLU versus the activity of the negative control (i.e., empty TF-expression vector). Luciferase data are based on six or ten independent biological replicates (i.e., a well of cells that were independently transfected with the reporter constructs, lysed, and prepared for measurement) from which outliers were excluded using the fourth-spread method. The standard error of fold change is computed for all TFs according to the rules of error propagation.

## MITOMI-based analysis of regulatory elements

MARE was performed according to the protocol developed for the MITOMI technology, including the design and fabrication of the molds for microfluidic devices and the devices themselves (Maerkl and Quake, 2007, 2009). To examine DNA elements using MARE, we first explored different strategies to divide long sequences into shorter fragments. Balancing overall throughput with the provided binding site detection resolution, we found that the most optimal library consists of 36 bp sequences covering the DNA element including 24 bp up- and downstream of the element so that each 36 bp fragment has a 24 bp overlap with the next one in the library, and that each 12 bp region is covered by three different fragments except for the first two and last two fragments which are covered only twice. We also flanked each fragment by a CCC clamp at the 5′ end to prevent degradation and by a complementary sequence at the 3′ end to allow primer hybridization. All fragments were obtained as single-stranded

oligonucleotides from Life Technologies. These oligonucleotides were subsequently used to generate labeled double-stranded DNA targets as described previously (Maerkl and Quake, 2007).

To enable the off-chip expression of TFs and their fluorescence-based detection, we again explored different strategies. We found that the WG *in vitro* transcription translation expression system containing translation enhancer (TE) sequences from the barley yellow dwarf virus (BYDV) (Promega) yielded the most robust and reproducible protein expression. To make this expression system compatible with our mouse TF ORF clone resource and to allow the fluorescence-based detection of TFs, we generated a novel vector, pMARE (Supplementary Figure S11), by cutting the pF3A WG (BYDV) Flexi vector (Promega) with *Nco*I and *Dra*I, thereby removing the barnase cassette. The Gateway reading frame A cassette (Invitrogen) was ligated in by the use of blunt-end cloning. Subsequently, the eGFP coding sequence (EUROSCARF) containing a stop codon at the 3′-end was incorporated between the *Kpn*I and *Sac*I restriction sites using standard cloning techniques. TFs were then subcloned from the entry clones into the pMARE vector by standard Gateway cloning. Linear expression templates containing the 5′ and 3′ UTR sequences as well as the TF ORF fused with the eGFP coding sequence were generated by PCR. Primers for PCR were designed through the Primer3Plus web application (http://www.bioinformatics.nl/cgi-bin/primer3plus/primer3plus.cgi; Fw: GTATCCGCTCATGGATCTCGATC, Rv: CGGTTTTATGGACAGCAAGC GA). Briefly, the PCRs contained 5 pmol of each primer, 100 ng of expression plasmid, 200 μmol of each dNTP, and 1 unit of iProof High-Fidelity DNA Polymerase (Bio-Rad) in a final volume of 50 μl. The reaction was cycled for 2 min at 98°C, followed by 35 cycles of 98°C for 15 s, 60°C for 30 s and 72°C for 2 min, followed by a final step of 5 min at 60°C. Each PCR was performed four times after which the resulting PCR products were pooled, PCR purified, and subsequently used as a linear expression template in a MARE experiment (Figure 4A).

All labeled double-stranded oligonucleotides were spotted onto an epoxy-coated glass slide (CELL Associates) with a SpotBot3 micro-arrayer (Arrayit) using a 946MP4 pin (European Biotek Network SPRL). The spotting protocol was done as follows: target DNA in 0.5% BSA aqueous solution was spotted first. Thus, each resulting spot consisted of a specific DNA fragment of the regulatory element (Figure 4A). TF proteins were synthesized by the use of the TnT SP6 High-Yield Wheat Germ Protein Expression System (Promega). The device alignment, surface chemistry, MITOMI, and data extraction were performed according to Maerkl and Quake (2007), except that an anti-GFP antibody was used to pull down the tagged TFs and that a multiplexer design was implemented to allow analysis of off-chip-expressed TFs (Rockel *et al*, 2013). For each 12 bp region, the average signal $S$ of the three fragments in which it is represented was determined. For each 12 bp region, we defined the mid position as the representative binding event position. Signal values between representative binding event positions were approximated by cubic interpolation (*interp1* function, signal package, R). Specific TF–DNA interactions were detected by clustering the signal of each position into two distinct classes, that is, specific binding positions (SBPs) and non-specific binding positions (NSBPs), using the *k*-means clustering algorithm (*kmeans* function, R; settings: centers = 2, algorithm = Hartigan-Wong, nstart = 1000). We subsequently defined the center of the NSBP class as the DNA bait-specific mean background signal (MBS). For each SBP, we defined the relative enrichment over non-specific binding as $E(SBP) = S(SBP)/MBS$ and removed SBPs with an $E$ of <2. Specific binding regions (SBRs) were detected by grouping consecutive SBPs, and SBR maxima were defined as the SBPs with the largest enrichment within an SBR. Each MARE experiment was independently reproduced. We defined a peak as reproducibly detected when the SBR maximum of the first replicate overlapped with the SBR of the second replicate and vice versa. For visualization, we scaled the *y* axis in Figures 4 and 5 and Supplementary Figures 12, 13, 16 by dividing the DNA occupancy signal of each 12 bp DNA element by the maximal signal observed across all 12 bp DNA elements.

### *In silico* prediction of TF–DNA interactions

DNA baits were scanned for putative TF–DNA interactions using the FIMO motif scanning algorithm (Grant *et al*, 2011) using three different stringency settings ($P < 1e-3$, $P < 1e-4$ (default), and $P < 1e-5$). PWMs were obtained from Jolma *et al* (2013). This data set contains 830 binding profiles for both human and mouse TFs as well as for full-length TFs and TF DBDs only. We therefore first curated this PWM data set to have an unambiguous representation of the binding specificities for our mouse TF clone collection. *First*, we only considered PWMs derived from full-length proteins for cases where both the full-length proteins and DBDs were probed. *Second*, we only considered mouse PWMs when data for both the human and mouse TF versions were available. *Finally*, when PWM information was available for human TFs only, we assessed the human-mouse orthology status and only considered PWMs derived from one-to-one orthologs as assessed by ENSEMBL (type 'ortholog_one2one'). For the MARE binding site analysis, we also retrieved PWMs from TRANSFAC (www.gene-regulation.com) for the TFs SMAD4 (V\$SMAD4_Q6), ETS1 (V\$ETS1_B), and ETS2 (V\$ETS2_Q6). TF binding sites for MARE occupancy profiles were predicted with a *P*-value threshold of $P < 1e-3$. To benchmark our Y1H assay, we calculated the PPV (or precision rate), that is, the fraction of Y1H TF–DNA interactions that are supported by a motif hit, and sensitivity, that is, the fraction of motif hits detected by Y1H, for different Y1H interaction sets (Supplementary Table S4).

### Nucleosome position analysis

We scanned all seven DNA baits for nucleosome positions with the Kaplan *et al* (2009) model (Supplementary Figure S9). We used the online version with default settings and 1 kb flanking regions from the pMW2 vector to account for boundary effects (http://genie.weizmann.ac.il/software/nucleo_prediction.html). We arbitrarily defined DNA positions as having a low-to-high (*P*-occupied > 0.3), moderate-to-high (*P*-occupied > 0.5), and high probability (*P*-occupied > 0.7) for nucleosome occupancy. Y1H PPV and sensitivity scores were calculated for DNA regions that were not occupied by nucleosomes at a respective occupancy probability. Specifically, if at least one residue of a binding site has a nucleosome probability above a certain threshold, then this site is no longer considered 'detectable' by Y1H.

### ChIP assays in NIH-3T3 cells

NIH-3T3 confluent cells were incubated overnight in serum-free medium and were stimulated with or without 10 ng/ml mouse TNF-α for 3 h (Life technologies). The cells were fixed as described previously (Raghav *et al*, 2012) and stored at −80°C. Five million cells were used for each immunoprecipitation (IP) and were lysed and sonicated as described previously (Raghav *et al*, 2012). After having estimated the chromatin concentration using a NanoDrop, an equal amount of chromatin for a control IP (rabbit control IgG) and specific antibody IP was taken. The chromatin sample was adjusted to 1 ml for each IP with the ChIP dilution buffer (1.2 mM EDTA pH 8.0, 16.7 mM Tris–HCl pH 8.0, 1% Triton X-100, 167 mM NaCl and 0.1% SDS containing protease and phosphatase inhibitors) after which the chromatin sample was incubated overnight at 4°C with each respective antibody (SP3: sc-644; SP4: sc-645; NFKB1: 50% ab7971 and 50% SC-372; 5 μg per IP) and a rabbit control IgG (Santa Cruz, sc-8994) coupled to magnetic sheep anti-rabbit IgG Invitrogen Dynabeads, as described (Myers Lab ChIP-seq Protocol, v041610.1 and v041610.2; http://www.hudsonal-pha.org/myers-lab/protocols/) with few modifications. Specifically, 50 μl of antibody-coupled beads was added to each 1 ml chromatin sample instead of 100 μl. After incubation, the beads were washed five times with a LiCl wash buffer (100 mM Tris at pH 7.5, 500 mM LiCl, 1% NP-40, 1% sodium deoxycholate) through mixing for 10 min at 4°C after which a 1-min mix at 4°C was used to wash the beads with 1 ml of TE (10 mM Tris–HCl at pH 7.5, 0.1 mM Na$_2$EDTA). The beads were then re-suspended in 200 μl IP Elution Buffer (1% SDS/0.1 M NaHCO$_3$), incubated in a 65°C shaker for 1 h and placed on a magnet to recover the supernatant. The supernatant was incubated at 65°C overnight to complete the reversal of the formaldehyde crosslinks. The next day, DNA was purified from the reverse-crosslinked chromatin using proteinase and RNase digestion as well as Qiagen DNA purification columns. The purified DNA was eluted in 30 μl of Qiagen elution buffer

and stored at $-20°C$ until verification of ChIP enrichment by qPCR. The following sense and antisense primers for the *Mmp9* promoter were used: Fw_TCTTTCCTTCCCCAAGGAGT and Rv_CCATCCCCACAC TGTAGGTT. One intergenic genomic region was chosen as a negative control as also described in Raghav *et al* (2012) with the following primers: Fw_CACACAGCTGACCTCCAGAA and Rv_ AGTGGCAAGGTC TCTGCTTC. The fold change between the bound *Mmp9* promoter and the negative control was calculated for each IP.

## Lentiviral vector mediated transgenesis

We cloned the *Mcts2-Id1* enhancer element in a *LacZ* reporter lentiviral vector construct containing a minimal promoter (gateway pRRLβLac, Friedli *et al*, 2010; Delpretti *et al*, unpublished). The lentivirus productions and the injections were performed as described in Friedli *et al* (2010). After the lentiviral vectors were injected in fertilized oocytes, we harvested mouse embryos at embryonic day 10.5 and observed enhancer expression by X-gal staining, according to standard protocols. The injection efficiency was determined by qPCR, after extracting the DNA from the yolk sac using a DNeasy blood and tissue kit according to the manufacturer's recommendations, as also described previously (Barde *et al*, 2011).

## Supplementary information

## Acknowledgements

We thank J Reece-Hoyes and M Walhout for constructive discussions regarding the Y1H assay workflow; S Maerkl, S Delpretti, and D Duboule for helpful discussions regarding, respectively, the MITOMI Technology and the *in vivo LacZ* reporter assay; G Udin for his help in the ChIP experiment; and the members of the Genomic Technologies Facility (GTF) and the transgenic core facility (TCF) for their help with the high-throughput sequencing (GTF), and the lentiviral-mediated transgenesis work (TCF). This work was supported by funds from the Swiss National Science Foundation, by a Marie Curie International Reintegration Grant (BD) from the Seventh Research Framework Programme, by SystemsX.ch (CycliX and the iPhD Program (JS)), by the NCCR Frontiers in Genetics Program, by the Japanese-Swiss Science and Technology Cooperation Program (ETHZ), and by Institutional support from the Ecole Polytechnique Fédérale de Lausanne (EPFL).

*Author contributions:* BD supervised the study. CG and BD designed the study. CG, JS, and BD built the TF clone collection. CG, A Iagovitina, JD, and KH performed Y1H screens. SKR and CG generated the high-throughput sequencing libraries. CG performed the luciferase assays. A Isakova and WH performed the MARE experiments. SMW performed MARE, DNA sequence, and Y1H data analyses. CG, SMW, and BD wrote the manuscript.

## Conflict of interest

The authors declare that they have no conflict of interest.

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

**The 2nd**
**International Conference**
**on Genomics in the**

# Americas

**September 12 - 13**
**Sacramento, USA**

Co-organizer : UC DAVIS

# The 8th

**International Conference**
**on Genomics**

**October 30 - November 1**
**Shenzhen, China**

Co-organizer : GigaScience

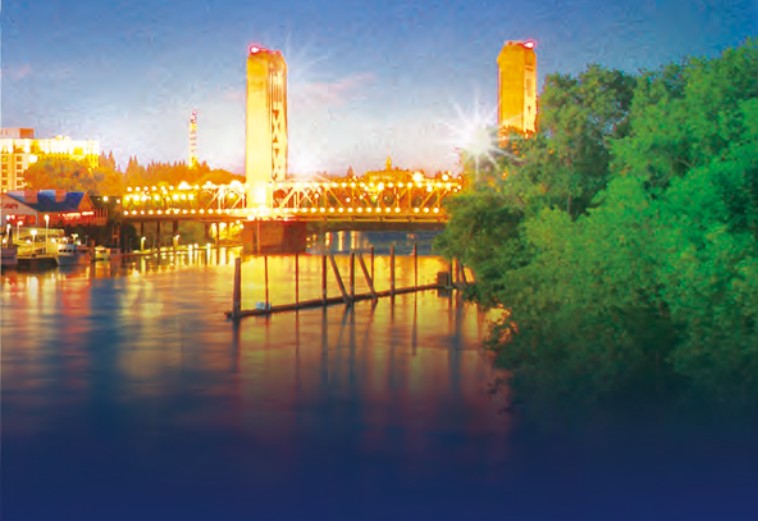

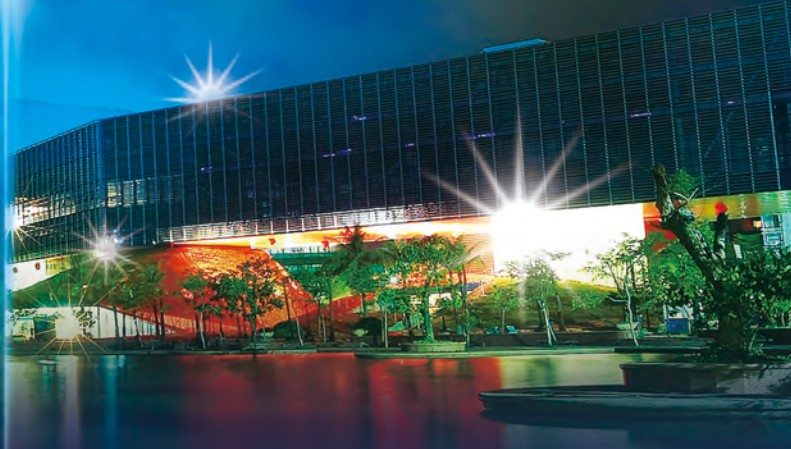

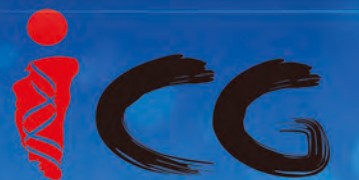 **ICG** | **Join Us for 2013**
**International Conference on Genomics**

Over the past seven years, the International Conference on Genomics (ICG) has been one of the top grade gathering of global thought leaders in genomics featuring latest advancements in genomic-related fields. This year, BGI continues to hold series ICG conferences, including ICG-8, ICG-Americas 2013, and ICG Europe 2013. These gatherings will be an excellent opportunity to exchange your research experience and latest discoveries, as well as the new insights into future development of life science.

**www.icg-2013.org**      bgi-event@service.genomics.cn      +86-755-25273340      Organizer : 华大基因 BGI