## [Review Process File · Molecular Systems Biology]

A yeast one-hybrid and microfluidics-based pipeline to map mammalian gene regulatory networks

Carine Gubelmann, Sebastian M. Waszak, Alina Isakova, Wiebke Holcombe, Korneel Hens, Antonina Iagovitina, Jean-Daniel Feuz, Sunil K. Raghav, Jovan Simicevic, Bart Deplancke

Corresponding author: Bart Deplancke, EPFL

Review timeline:

Submission date:	10 January 2013
Editorial Decision:	06 February 2013
Revision received:	04 May 2013
Editorial Decision:	09 June 2013
Revision received:	21 June 2013
Accepted:	28 June 2013

Editor: Thomas Lemberger

Transaction Report:

1st Editorial Decision

06 February 2013

Thank you again for submitting your work to Molecular Systems Biology. We have now heard back from the two referees who agreed to evaluate your manuscript. As you will see from the reports below, the referees find the topic of your study of potential interest. They raise, however, several concerns on your work, which should be convincingly addressed in a revision of this work.

Without repeating all the points listed by the reviewers below, issues that should be addressed include the following:

- the results should be analyzed in the light of the performance of motif scanning using high-quality PWM.
- a ChIP validation should be included.

If you feel you can satisfactorily deal with these points and those listed by the referees, you may wish to submit a revised version of your manuscript. Please attach a covering letter giving details of the way in which you have handled each of the points raised by the referees. A revised manuscript will be once again subject to review and you probably understand that we can give you no guarantee at this stage that the eventual outcome will be favorable.

 REFEREE REPORTS:

Reviewer #1

This paper has several offerings which will be of interest to the field:

- (1) A new collection of mouse TF clones, which includes sequence-verified versions of the majority of mouse TFs (750).
- (2) Comparison of outcomes from screening the TF library against seven known or putative mouse enhancers, using a mating-based system and a transformation-based system.
- (3) Followup using the MARE assay.
- (4) Some in vivo analysis verifying the importance of identified regulatory sites.

The paper is intended to showcase the pipeline, and while it does have some biology in it, it focuses on technical issues. While I do see this as important work, the paper in its current form strikes me as less than the sum of its parts, and I have several issues with the claims made in the analyses. I believe that all of this can be solved with data analysis and writing. The key issue is whether there really is a lot of in vitro binding that cannot be explained with motif matches, since the assays used here would be able to pick it up. In fact, it is important for this paper that PWMs are erroneous, because if they were accurate, then the two techniques coupled here would be more or less redundant with PWM scans. However, the idea that PWMs are so disastrous seems counter to conventional wisdom (and much of the recent literature on the topic). The one paper cited in this regard is from eight years ago and it isn't really a good reference for this (see below).

In essence, the whole PWM scanning issue needs to be re-examined, and after that, the benchmarking, performance, and motivation for the system described here need to be revisited in comparison to PWMs scans, which the Introduction already points out are the obvious thing to do. It would help to show that at least one of the protein-DNA interactions detected wouldn't have been predicted from a PWM scan. For example, the ones that were followed up with biological assays.

It would also be helpful to give some estimate about the throughput of these assays. How long does it take to look at ten enhancers? There seem to be hundreds of thousands in human and mouse; is it possible for the assays to be scaled up that much in the foreseeable future?

Major comments:

1. A key issue is presentation and ordering of information. It is known that Y1H and Y2H assays have both low true-positive rates and high false-positive rates (the latter seems to be less of a problem than the former in this study). The Discussion makes some arguments about these things, but I think the Introduction needs to lay it out. Why should we use these approaches? What exactly do they provide that we couldn't get from scanning with motifs, and why are they able to provide it? What is the unique advantage here? I think it comes down to whether or not PWMs or other representations are sufficiently accurate, but this discussion is given short-shrift - see below.

2. The section on "benchmarking" could be much more clear - I had to read it several times before I understood what had been done and why. It is not mentioned until the end that there are known positives; certainly that should be mentioned at the beginning. Discussion of the relatively high false-positive and false-negative rates associated with Y2H and/or Y1H should also be discussed at the outset, so that the experimental design makes more sense. It might be useful to use motif matches as a benchmark; so many mouse TFs have a motif and it is easy to scan for binding sites.

3. The fact that the recall rate is only 18% overall is surely something that most readers would want to know more about - the paper should perhaps at least put forward some potential explanations for that. In fact, overall, it seems that the Y1H hit rate is very low; only 7 to 13 out of 750 screened. This number seems inconsistent with the fact that most mouse TFs have fairly small and degenerate PWMs, so that on average there would be a good binding site every few kb. If the elements tested are on the order of a kb, then shouldn't there be a lot more hits? In fact, shouldn't it be possible to calculate what hits to expect, given that there are now PWMs for hundreds of mouse and human TFs (and more, if one considers paralogs)? Although the paper puts forward problems with PWMs as a source of discrepancy, I am actually guessing that the explanation may be that much of the enhancer ends up occluded by nucleosomes when it is put into yeast, so a lot of the sites are not accessible. Yeast will essentially "chromatinize" anything unless it has high A/T content, or poly-A or poly-G

stretches. It would be worth asking whether success or failure to get a Y1H hit when there is a perfect motif match can be explained by local sequence context. There are several ways to predict nucleosome occupancy from sequence - the Kaplan model, for example, is available online.

4. I don't buy that "our data illustrate how motif prediction by itself is not sufficient to resolve the precise binding pattern of TFs as has been previously observed (Vavouri & Elgar, 2005)"; which is stated after describing the fact that there are two NFkB1 motif matches in the Mmp1 promoter which show little to no binding in the MARE assay. There are several issues with this statement and I believe it may be erroneous. It is really at the core of the need for this whole system, so it's important to get it right here.

First, Vavouri and Elgar discussed prediction of functional regulatory sequence *in vivo*; this is an entirely different ball game from *in vitro* binding, which should be just biochemistry with protein vs. DNA. So, the reference is inappropriate, as what is being considered here is *in vitro* behavior.

Second, the more recent literature on this topic (Zhao and Stormo, NB 2011; Weirauch et al., NB 2013) indicates that when PWMs fail it is usually because the PWM does not really reflect the biochemical activity of the protein - you can't expect a bad PWM to be a good predictor. So, it is possible that the problem here is not that a PWM wouldn't work, but that the PWM being used does not accurately reflect the binding of the protein to different sequences.

This needs to be explored. Where did the PWMs used come from? Were other PWMs tried? Is it really the same protein being tested? How was the thresholding done? Why not show a scatter plot of PWM score vs. MARE score, instead of the linear and thresholded versions used? What sequences underlie these motif matches that are not bound - are they different from the ones that are bound in some way? They should be shown in the figure. They are apparently not consensus matches. And what is "consensus", exactly? Is that a match to the IUPAC-ized version of the PWM?

The most critical issue is to be using the best PWMs. The methods state that TRANSFAC and JASPAR motifs were used, but these databases are highly heterogeneous in quality, so the motifs may not be the best available. The 2011 Wong et al. paper "Extensive characterization of NF- κ B binding uncovers non-canonical motifs and advances the interpretation of genetic functional traits" should be investigated - it much more carefully explores the sequence preferences of NFkB family members. The paper "Genome-wide analysis of ETS-family DNA-binding *in vitro* and *in vivo*" carefully considers the motifs for ETS family members, showing that most of the previous motifs (i.e. the ones in Transfac and Jaspar) are probably wrong. ZIC3 and its paralogs have motifs in several papers. I believe the others may as well - this needs to be looked into.

And, the sequences that are bound but don't have a PWM match need to be shown. An alternative possibility is that the MARE assay has a systematic false-positive issue. That might become clear by looking at the sequences.

Reviewer #2 (Remarks to the Author):

Gubelman et al. "A yeast one-hybrid and microfluidics-based pipeline to deorphanize mammalian gene regulatory elements." Submission to Molecular Systems Biology.

In this manuscript, Gubelman et al report a systematic and high-throughput approach to identify and validate which mouse TFs bind to specific regulatory regions. Extending similar approaches the Deplancke group has taken with great success in *Drosophila* (Hens et al 2011), now in mouse they have created a large-scale ORF collection of over 700 TF cDNA sequences that is employed in two approaches, one mating-based and one 1-hybrid based. They then validated their results obtained against nine promoter/enhancers using luciferase and MITOMI based analyses.

Overall, this is a solid re-deployment of their prior approach used in *Drosophila*. It is impossible not to note that this MSB study is mostly structurally cloned from the Nat Methods paper in 2011. Importantly, however, they have opened what may well become a major new frontier in transcriptional investigation by demonstrating the usefulness of this approach to mammalian systems. Direct experimental tools that are conceptually reverse-ChIP experiments are very much needed to complement the one-directional fishing expeditions undertaken by so many groups, such

as ENCODE.

I have a single major suggestion for their study, a suggestion purely calculated to maximize its impact across the highly competitive and crowded field of mammalian transcription. Gubelman et al ought to take their validation strategy beyond what has been previously reported (e.g. Hens 2011), and specifically add ChIP experiments as a third confirmation. Pragmatically, this would greatly increase the appeal of their approach to groups such as the ENCODE crowd, since chromatin IPs have become so much a focus of systems biology. I understand the counter-arguments (referenced as Deplancke 2009), and I am also aware that this is stretching them into a new technical area, but if they bite the bullet on adding at least one validating ChIP experiment, it would enormously improve the paper's impact and appeal.

Minor points:

- 1) In the Title: 'deorphanize' sounds like an urgent surgery. Perhaps the title could be re-worded?
- 2) I think that Stamatoyannopoulos 2012 should be replaced with the main ENCODE paper, which is doi:10.1038/nature11247
- 3) The last three sentences of the intro are awkward, and could use editing.
- 4) 'Quasi hands-off screening' phrase is awkward.
- 5) The justification of their chosen targeted promoters and enhancers (top of page 6) could (and should) be more clearly justified and more slowly elaborated. Across this manuscript, it often feels like more time is spent discussing technical aspects, and less on the targeted biology. Both tech and bio should feature highly in an MSB paper.
- 6) (page 6) "greater number of interactions that were" I think they meant to say '...than were' Continuing right after this phrase, I suggest reorganizing the following two to three sentences, which were difficult to parse.
- 7) (page 6) Last sentence uses 'thus' twice. Both can be deleted, I think.
- 8) (page 7: twice) Ending many sections with a sentence starting "Together,..." can be useful as a summary, but I would prefer more eloquent sentences than are used on page 7. The current ones are a bit clunky.
- 9) (page 8) The description of 'on and off chip produced TFs' was also difficult to follow.
- 10) (page 10) "In vivo validation of detected TF-enhancer interactions" would be greatly strengthened by doing one or more ChIP experiments (per the main comment). I entirely understand and sympathize with the counter-arguments referenced in Deplancke 2009, but I think the time has come to add these direct, in vivo protein-DNA contacts as a third line of validation.
- 11) The phrase 'as such' and the word 'thus' are used a lot in this paper, and can likely be deleted almost everywhere.
- 12) (page 14) In the discussion, there are a few uses of quotations to describe things like "known interactors" and "gold standard". These sections should likely be re-worded to remove these quotes, which should be used very sparingly if at all in formal scientific writing.
- 13) It looks like there are two Hens references that are identical, one should be deleted.
- 14) Figures 1 & 7 could be combined.
- 15) Figure 3 has a relatively low information density, and could be greatly compressed or even eliminated in favour of in-text description.
- 16) Many of the supplemental pdfs were not carefully printed from the original excel files, and significant text was lost from them.

Reviewer #1:

The reviewer points out that this paper is an “important work” and has “several offerings which will be of interest to the field”: the new collection of mouse TF clones, the comparison of outcomes using mating-based system and transformation-based system, the follow-up using the MARE assay and the *in vivo* analysis verifying the importance of identified regulatory sites.

>We thank the reviewer for pointing out the relevance of our work for the gene regulation community.

The reviewer raises however several concerns, which she/he indicates can be solved through data re-analysis and manuscript rewriting. For clarity, we have summarized the comments below:

1) The principal concern is about how well the combined Y1H and MARE pipeline complements PWM scans in detecting protein-DNA interactions with regulatory elements. In other words, the reviewer wonders whether there “really is a lot of *in vitro* binding that cannot be explained with motif matches (since the assays used here would be able to pick it up) as this would truly demonstrate the value of the proposed experimental pipeline. The reviewer therefore asks to re-examine the PWM scanning issue and to make sure that the best available motifs are used by also scanning the literature next to mining TF motif databases. Specifically, the reviewer suggests to:

- a) calculate how many hits one could expect based on scanning of all available PWMs and to discuss this finding in the main manuscript;**
- b) assess whether the relatively low true positive rate (18%) may be caused by the fact that “much of the enhancer ends up occluded by nucleosomes when it is put into yeast, so a lot of the sites are not accessible.” To test this, the reviewer suggests to predict nucleosome occupancy from sequence (*e.g.*, using the Kaplan model) and to assess whether nucleosomes indeed mask many of the motifs for known positives that were not detected;**
- c) update the literature that is cited when discussing PWM scanning-related issues.**

>We thank the reviewer for her/his useful suggestions as these analyses indeed turned out to be valuable to further characterize the presented pipeline and results. However, before we were able to more comprehensively compare and contrast motif scanning results with those from the Y1H and MARE assays, we had to solve a couple of basic analysis issues:

First, and as the reviewer acknowledges, a TF may be associated with several PWMs, which may vary in quality dependent on the method by which they were derived. For consistency, we therefore decided to use a recently published high-throughput SELEX data set by Jolma *et al.* (Cell, 2013) because it contains the largest collection of high-quality PWMs for human and mouse TFs to date. We were able to retrieve reliable PWM data for 326 out of the 750 TFs (43%) within our clone collection, based on the fact that these TFs were directly probed or at least their one-to-one human orthologues. The absence of PWM models for the majority of TFs in our collection emphasizes however the utility of the Y1H assay in that it enables the screening of many uncharacterized TFs in relatively unbiased fashion.

Second, motif scanning methods differ in accuracy and moreover appear to apply different thresholding and background filtering rules. To circumvent this issue, we selected a widely applied method in the field, FIMO, which is part of the well-known MEME software package and which was shown to compare favorably to other motif scanning tools in terms of search functionalities (Grant *et al.*, Bioinformatics, 2011). We scanned the seven bait sequences for the presence of motifs at three different detection thresholds including the default parameters to benchmark the performance of the Y1H and MARE assays.

Our analyses revealed that using default motif scanning parameters, ~50% of reproducibly detected Y1H interactions are supported by respective motif hits (see the “all” category in revised **Supplementary Table S4**). The precision rate or positive predictive value (PPV) increases to ~90% when motifs are called at lower stringency, thus providing computational DNA binding support for the vast majority of reported interactions. We did not detect any clear emerging pattern regarding which Y1H procedure is most robust, although consensus interactions (*i.e.*, interactions reproducibly detected by both transformation and mating) tend to have the highest PPV value at high motif detection thresholds. This indicates that interactions that are driven by binding sites that closely resemble the consensus sequence will likely be detected by both Y1H approaches. Interestingly, the PPV values for our positive control interactions (*i.e.*, those reported in the literature) were only slightly higher than all the Y1H ones (see revised **Supplementary Table S4**), indicating that reproducibly detected Y1H- and literature-based interactions feature a comparable probability in respective motif presence. A similar trend was observed for interactions that were positive in luciferase reporter assays. In addition, 16 out of the 40 detected Y1H interactions involving 10 TFs were not predicted through motif scanning. For two TFs, PWMs are available within the Jolma *et al.* (Cell, 2013) data set, yet motifs were not predicted within the respective DNA baits (the *Fos* promoter and *Mlcrhs4* element for the TF BARX1 and the *Mcts2-Id1* enhancer for the TF ONECUT2). A further scan of other databases (*e.g.*, TRANSFAC, JASPAR, UNIPROBE) revealed that PWMs may be available for five out of the remaining eight TFs, leaving three TFs (CDX4, NR2C1, ZDHHC9) involved in four interactions uncharacterized. This illustrates the complexities of PWM scanning and again emphasizes the utility as well as complementary nature of Y1H screens.

Response to question “a” above: Many more motifs were however called for which the respective TF-DNA interactions were not detected using Y1H. This is revealed when calculating the sensitivity, which represents the percentage of motif-predicted interactions supported by Y1H. We found that the sensitivity amounts to ~2%, ~3%, and ~10% of all reproducibly detected interactions for respectively low ($P < e^{-3}$), medium (default, $P < e^{-4}$), and high ($P < e^{-5}$) motif calling stringencies (see the “all” category in revised **Supplementary Table S4**). These numbers likely reflect on the one hand the fact that motif scanning may be prone to false positive calls (Medina-Rivera *et al.*, 2011; Klepper and Drabløs, BMC Bioinformatics, 2013), but on the other the likely substantial technical false negative rate of the Y1H assay. One possible reason for the high false negative rate and by extension the low positive detection rate of the Y1H assay may be the occlusion of binding sites by nucleosomes, as suggested by the reviewer. In other words, since the DNA baits are chromatinized, genuine binding sites may not be accessible because of nucleosomal interference.

Response to question “b” above: To test this, we predicted the nucleosome occupancy of each DNA bait using the Kaplan *et al.* model (Nature, 2009). All seven DNA baits showed an overall high predicted nucleosome occupancy landscape (**Supplementary Figure S9**), which suggests that many interactions which involve sites that are predicted to be strongly occupied by nucleosomes are nevertheless detected by Y1H. An example is the interaction between the *Mcts2-Id1* enhancer and RFX2 whose motif is predicted to be located in a high nucleosome occupancy ($P>0.7$) region. This observation is consistent with the PPV values, which gradually decrease upon progressively excluding regions within DNA baits that feature a strong ($P>0.7$), moderate ($P>0.5$) or weak ($P>0.3$) predicted nucleosome occupancy (**Supplementary Table S5**). In other words, by removing regions within DNA baits with moderate to strong predicted nucleosome occupancy, we remove many motifs that support detected Y1H interactions. However, when calculating the corresponding sensitivity values, we observed an inverse relationship when considering all Y1H interactions supported by high stringency motifs. This suggests that sites that are predicted to be weakly occupied with nucleosomes and have a high motif score are more likely to be detected. Providing that the predicted data reflect true *in vivo* behavior, our analyses therefore do not point to nucleosome occupancy as the main factor that affects the false negative (or true positive) rate, although the most accessible consensus sites have clearly a greater detection likelihood.

To further explore the molecular basis for the probable high false negative rate of the Y1H assay, we examined whether the overall motif load for a specific TF within a DNA bait affects the interaction detection probability. As shown in revised **Supplementary Table S6**, we found that the sensitivity of the Y1H assay is indeed partially dependent on both motif number and score in that the higher both are, the more likely the respective interaction will be detected. We conclude that motif number, score, and nucleosome occupancy are all factors that clearly influence the Y1H detection and thus false negative rate.

We have included these new findings in the revised manuscript (p. 8-10)

d) show a scatter plot of PWM score vs. MARE score, instead of the linear and thresholded versions used to alleviate thresholding issues and to more closely contrast the motif matches that are bound from those that are not;

>We integrated the new motif scanning data with results from our MARE analyses. Specifically and as requested by the reviewer, we plotted the updated PWM versus the MARE scores of the tested interactions. As shown for example for SP3 and 4 (**Figure 5A** and revised **Supplementary Figures S14, S15**), the MARE scores follow the motif score landscape well although the correlation deteriorates for very low scoring motifs, many of which are therefore likely false positives. This indicates that our MARE assay is able to accurately capture the DNA binding behavior of the majority of tested TFs. Or reversely, that PWM scores are good, quantitative read-outs of the possible binding sites of a TF, as observed earlier (Wasserman and Sandelin, Nature Reviews Genetics, 2004; Stormo and Zhao, Nature Reviews Genetics, 2010; Weirauch *et al.*, Nature Biotechnology, 2013). Our data is therefore consistent with the observation that TFs can bind to sites that substantially diverge from their consensus sequence (*i.e.*, they have low motif scores), owing to the degenerative nature of TF sequence specificities (Weirauch *et al.*, Nature Biotechnology, 2013).

We have included these new findings in the revised manuscript (p. 13).

e) show that at least one of the protein-DNA interactions detected wouldn't have been predicted from a PWM scan. For example, the ones that were followed up with biological assays;

f) show the sequences that are bound but don't have a PWM match.

>Upon mining our MARE data, we found several instances involving the TFs ONECUT2, ETS1, and ETS2 where the binding peak of the TF does not correspond to a predicted motif. Among these, ONECUT2 is perhaps the most interesting because its interaction with the *Mcts2-Id1* enhancer has been validated using enhancer deletion constructs as well as luciferase reporter assays. Moreover, the biological relevance of this interaction was supported by our *in vivo* enhancer activity experiments. We have listed the sequences through which these TFs likely interact in revised **Supplementary Table S7**, but a more detailed molecular dissection of the implicated DNA sequences and protein domains will be required to conclusively define these sequences as alternative or secondary binding sites of these TFs. For example, we observed that a reproducibly detected ETS1 peak is not supported by a motif call (**Figure 4C** and **Supplementary Figure S13**). Upon closer inspection, we found that an ETS-like binding site is present, but its score ($P=1.2e-3$) fell just below our low detection threshold of $P<1e-3$. Intriguingly, when the underlying PWM was replaced by a model from TRANSFAC, then the motif was scored as significant ($P=2.2e-4$), illustrating the difficulty in obtaining an accurate snapshot of the DNA binding landscape purely based on motif calling. We have included these results in the revised manuscript (p. 13-14).

2) The reviewer indicates that it would be “helpful to give some estimate about the throughput of these assays. How long does it take to look at ten enhancers? There seem to be hundreds of thousands in human and mouse; is it possible for the assays to be scaled up that much in the foreseeable future?”

>Our pipeline has two throughput aspects: on the one hand, both the Y1H and MARE assays allow the screening of more than 700 protein-DNA interactions in one experiment, which can, in our opinion, be considered “high-throughput” or “large-scale”, hence our use of this terminology. On the other hand, each experiment requires the preparation of respective yeast bait strains or microfluidic chips, which dampens the true “throughput” of the pipeline. The latter goes to the heart of the reviewer’s question as it directly affects the capacity of our pipeline to screen many elements in a relatively short amount of time. Based on our experience, we believe that it is reasonable to suggest that at least 30 elements can be screened per month per person using the Y1H assay, whereas this is lower (~10) for the MARE assay. As such, we do not believe that our pipeline will ever be suitable to screen all predicted regulatory elements in the mouse or human genomes. Rather, we believe that our pipeline will be most useful for targeted experiments, screening elements that are active in particular biological processes or associated with specific types of genes. We have more clearly discussed this issue in the revised manuscript (p. 8).

3) The reviewer requests to better explain the true value of the proposed pipeline, and thus to better motivate why one would be interested in using these approaches, especially compared to *in silico* tools that allow motif scanning.

>*First*, our pipeline allows the screening of virtually every TF while motif scanning is limited to those TFs of which the DNA binding specificities (PWMs) have been characterized. For example, 10% of our detected Y1H interactions involved TFs for which to our knowledge no PWM is available.

Second, motif scanning is more complex than may seem at first sight. For example, the quality of PWMs is variable and several PWMs may be associated with the same TF, making it difficult to compile the most accurate or comprehensive data set. Perhaps even more importantly, we found that using default motif scanning parameters, about half of the detected Y1H interactions would not have been identified. One option would be to lower the required motif score because almost all detected Y1H interactions were supported by motifs detected with a lower stringency and because the MARE and motif scores correlate well. However, lowering the detection threshold increases the motif overcalling risk, resulting in a higher false positive rate.

We found that the sensitivity of the Y1H assay is ~3% at a default motif calling stringency. This means that about 60 reactions are predicted per DNA bait when the latter are scanned for motifs using a default threshold. While the Y1H assay has an important false negative rate, it is unlikely that all of these interactions are plausible from a molecular or technical point of view. Indeed, it would imply that 20% of all TFs (60/326) within our current Y1H assay whose binding specificities have been identified can interact with any given ~1,000 bp DNA sequence and that more than 40% of the screened DNA sequence would be involved in TF-DNA interactions (*i.e.*, based on motif scanning), which appears excessive.

Third, the points discussed above highlight the need for experimentally validated motif predictions, which the Y1H and MARE assays readily provide. A reasonable strategy therefore may be to combine motif calling with Y1H screens to yield the most comprehensive and robust set of protein-DNA interactions. These can then be further characterized using the MARE assay, highlighting the unique and complementary nature of the presented pipeline compared to motif calling.

We have integrated this response into the discussion of the revised manuscript (p. 17-18).

4) The reviewer requests to clarify the section on "benchmarking". Specifically, the reviewer suggests to discuss "the relatively high false-positive and false-negative rate associated with Y2H and/or Y1H at the outset", so that the experimental design makes more sense.

>To clarify the selection of promoter/enhancers and positive controls, we updated the text in the revised main manuscript (p. 7) and in the **Materials & Methods** section entitled "Selection of the positive controls for DNA bait interactions" (p. 24).

Reviewer #2 (Remarks to the Author):

The reviewer indicates that our paper “opens a major new frontier in transcriptional investigation by demonstrating the usefulness of this approach to mammalian systems” and that the pipeline “is very much needed to complement the one-directional fishing expeditions undertaken by so many groups, such as ENCODE”.

>We thank the reviewer for her/his enthusiasm for our work and for pointing out the relevance of our work to the scientific community.

Major issue: The reviewer suggests including at least one ChIP validation to improve the paper’s impact and appeal.

> To include one ChIP validation experiment of detected TF-DNA interactions, we first examined the availability of antibodies for all the 25 detected Y1H interactors (listed in **Figure 2**). Only for 10 out of these did we find antibodies that were either used in published ChIP experiments or tagged as “ChIP grade” by biotech vendors. Unfortunately, RFX2 or ONECUT2 are to the best of our knowledge not among these 10 TFs, preventing us to validate the detected interactions with the enhancer *in vivo*, although we suspect that this in any case would have been difficult given the arguments formulated before (*e.g.*, Deplancke (2009)). We therefore redirected our attention to interactions that may be testable in a cell culture system. Based on the study of Balasubramanian *et al.* (2011) which showed that NFKB1 interacts with the *Mmp9* promoter in NIH-3T3 cells after TNF- α induction, we decided to use this experimental set-up to validate the detected TF-*Mmp9* interactions. In our Y1H-based system, we identified SP3, SP4, NFKB1, and ZIC3 interacting with the *Mmp9* promoter. Given the antibody unavailability for ZIC3, we tested the interactions of SP3 and SP4, as well as NFKB1 as a control, to the *Mmp9* promoter by ChIP-qPCR, before and after TNF- α stimulation. As shown in the new **Supplementary Fig. S17**, SP3, SP4, and NFKB1 are binding more tightly to the *Mmp9* promoter in stimulated compared to unstimulated cells. We have included this new experiment in the revised manuscript (**p. 14-15**).

Minor points related to manuscript structure and writing:

>We thank the reviewer for her/his constructive comments to improve our manuscript. We have addressed all remarks as indicated below:

1) The reviewer indicates that the “deorphanize’ sounds like an urgent surgery” in the title.

>Taking into account the maximum character limit, we propose the following new title: “A yeast one-hybrid and microfluidics-based pipeline to map mammalian gene regulatory networks”.

2) I think that Stamatoyannopoulos 2012 should be replaced with the main ENCODE paper, which is doi:10.1038/nature11247

>We would like to point out that in the main manuscript, we typically refer to the *mouse* ENCODE paper (Stamatoyannopoulos, 2012) and not to the *human* ENCODE one. However, in the first sentence in the introduction, we have included the general reference as suggested by the reviewer (The ENCODE Project Consortium, 2012).

3) The last three sentences of the intro are awkward, and could use editing.

>We rephrased this section as follows: “MARE enables the simultaneous monitoring of more than 700 protein-DNA interactions, rendering this technology ideal to systematically scan relatively long regulatory elements for DNA occupancy of specific TFs. MARE is therefore a complementary approach to the Y1H assay because it can resolve individual TF binding sites

within targeted DNA elements (Simicevic & Deplancke, 2010). Thus, our Y1H and MARE methods constitute a powerful cross-platform pipeline to systematically characterize mammalian regulatory elements of interest for interacting TFs at unprecedented throughput and resolution”

4) 'Quasi hands-off screening' phrase is awkward.

>We deleted “quasi” for simplification.

5) The justification of their chosen targeted promoters and enhancers (top of page 6) could (and should) be more clearly justified and more slowly elaborated. Across this manuscript, it often feels like more time is spent discussing technical aspects, and less on the targeted biology. Both tech and bio should feature highly in an MSB paper.

> As also per reviewer’s one request, we updated the text in the revised main manuscript (p. 7) and in the Materials and Methods section entitled “Selection of the positive controls for DNA bait interactions” (p. 23).

6) (page 6) "greater number of interactions that were" I think they meant to say '...than were' Continuing right after this phrase, I suggest reorganizing the following two to three sentences, which were difficult to parse.

>We re-phrased this section in the revised manuscript for improved clarity (p. 7).

7) (page 6) Last sentence uses 'thus' twice. Both can be deleted, I think.

>We agree with the reviewer and removed “thus” in both occasions.

8) (page 7: twice) Ending many sections with a sentence starting "Together,..." can be useful as a summary, but I would prefer more eloquent sentences than are used on page 7. The current ones are a bit clunky.

>We rephrased this section in the revised manuscript (p. 8)

9) (page 8) The description of 'on and off chip produced TFs' was also difficult to follow.

>To improve clarity, we rephrased this section as follows: “In this new strategy, each tagged TF is first *in vitro* expressed in a microcentrifuge tube (off-chip) and then transferred in each specific chip row to test for binding to the spotted DNA sequences (**Figure 4A**). The advantage of off-chip synthesis is that it yields a larger protein amount relative to on-chip protein expression, which we found increases the overall signal-to-noise ratio and improves the binding site resolution (**Supplementary Figure S12**) (p. 12)

10) (page 10) "In vivo validation of detected TF-enhancer interactions" would be greatly strengthened by doing one or more ChIP experiments (per the main comment). I entirely understand and sympathize with the counter-arguments referenced in Deplancke 2009, but I think the time has come to add these direct, in vivo protein-DNA contacts as a third line of validation.

>Please see our response above to the reviewer’s main concern.

11) The phrase 'as such' and the word 'thus' are used a lot in this paper, and can likely be deleted almost everywhere.

>We thank the reviewer to point out this redundancy. We have removed or replaced most of these terms throughout the manuscript.

12) (page 14) In the discussion, there are a few uses of quotations to describe things like "known interactors" and "gold standard". These sections should likely be re-worded to

remove these quotes, which should be used very sparingly if at all in formal scientific writing.

>We have corrected this in the revised manuscript.

13) It looks like there are two Hens references that are identical, one should be deleted.

>We apologize for this oversight and corrected this in the revised manuscript.

14) Figures 1 & 7 could be combined.

>We respectfully disagree with the reviewer to remove the TF ORF cloning workflow from **Figure 1** and to combine it with **Figure 7**. This is because we believe that the mouse TF ORF library is an important part of the paper as it, given its format, should be of great value for the mouse community at large. Furthermore, the comparison of the mating and transformation approach is useful for the scientists who may not be familiar with these procedures. In **Figure 7**, we summarize the workflow of Y1H and MARE to assist the reader in understanding the flow of the presented pipeline. Both Figures therefore communicate a fundamentally different message and in our opinion should be kept separate.

15) Figure 3 has a relatively low information density, and could be greatly compressed or even eliminated in favor of in-text description.

>We compressed **Figure 3** by removing the schematic picture of the luciferase assay, and by reformatting the other panels.

16) Many of the supplemental pdfs were not carefully printed from the original excel files, and significant text was lost from them.

> In the final paper, the supplementary tables should be accessible in excel format only, which should alleviate PDF formatting issues.

Thank you again for submitting your work to Molecular Systems Biology. We have now finally heard back from the two referees who accepted to evaluate the revised study. As you will see, reviewer #2 is now fully supportive. Reviewer #1 feels that some revisions in the text and in the presentation of the data are still required to provide a more rigorous account of the results, in particular with regard to the data related to the impact of nucleosome occupancy. While this reviewer's report is very detailed, the issues can all be addressed with appropriate amendments and clarifications. It is nevertheless important that these points are taken very seriously in an exceptional second and last round of revision.

It seems also that error bars are provided in several figures even if the number of replicates is only of two (Fig 3B, 6A, supp fig S17, for example) and that technical and biological replicate are all considered as independent samples. For example the legend of Figure 3AB states "The error bars represent the standard error of two independent experiments with three replicates each (n=6)". Please correct the analysis and data presentation to follow our guidelines (see also <http://www.nature.com/msb/authors/index.html#a3.4.3>) : "Graphs must include clearly labelled error bars for cases where more than two independent experiments have been performed (error bars should not be shown for technical replicates)." and "For each experiment, the number of both technical and biological replicates should be clearly stated. Biological replicates are derived from independent experiments using separately obtained biological samples, while technical replicates are created by repeated measurements on the same biological sample. In general, technical replicates should be averaged before any statistical inference tests are performed". For these figures (reporter assays), please upload the Excel files (or csv) that include the individual values for technical and biological replicates as 'figure source data' associated with the respective panels (see also 'Source data for figures' under <http://www.nature.com/msb/authors/index.html#a3.4.3>).

REFEREE REPORTS:

Reviewer #1

The revised manuscript is much more convincing - the need for the reagents and methodology, its performance, and its relationship to alternatives are considerably more clear. I feel it will make an important contribution to the literature.

There are still a few key points that don't make sense though and they should be addressed prior to publication in my view - at best, they will be confusing; at worst, they are either wrong or highly misleading. I think these can be solved by rewriting and should not require re-review.

(1) It is still not clear how many "hits" were obtained from the Y1H screens - when figures are presented in the text, they should refer to specific rows and columns in the Tables, and particular care should be taken to ensure that information is presented in order. I found myself uncertain in many places how various figures were obtained. For example, on P. 7, it is stated that "We identified 40 protein-DNA interactions involving 25 distinct TFs and seven DNA baits." But, in the discussion of motifs, it says "In addition, 16 out of the 40 detected Y1H interactions involving 10 TFs were not predicted through motif scanning". I cannot reconcile these numbers.

(2) The statement that "16 out of the 40 detected Y1H interactions involving 10 TFs were not predicted through motif scanning" is immediately followed by the statement "A further scan of other databases (TRANSFAC (Wingender, 2008), JASPAR (Bryne et al, 2008), and UNIPROBE (Robasky & Bulyk, 2011)) revealed that PWMs may be available for five out the remaining eight TFs." So, most of the 16 interactions "not predicted" can be explained because the analysis omitted published data? Why were these motifs not included? Later, in the MITOMI section, it seems that they are considered. It seems that it should be fairly straightforward to incorporate these motifs earlier in the paper - a major conclusion of the paper is that the new methods provide something that

motifs don't, so it's important to give the motifs a fair shake.

(3) The following statement requires clarification: "motif scanning may itself be prone to false positive calls (Klepper & Drablos, 2013; Medina-Rivera et al, 2011)". I don't see what data in these papers supports this statement beyond the obvious fact (already noted in my previous review) that, *in vivo*, not all motif matches are occupied for reasons such as nucleosome occupancy, competition among TFs, chromatin state, etc, even though the TF would bind the site in a biochemical experiment with purified DNA and protein. The Y1H assay is supposed to detect protein-DNA interactions independently of chromatin (although clearly it does not really do that from the analyses presented here!) so the literature on motifs vs. DNA needs to be cited if the idea of motif models is going to be critiqued. I think the original papers from Bulyk, Stormo, Benos, etc and maybe some of the followup large-scale studies would probably be best - and it would be beneficial to the reader to know exactly why there would be false positive calls. Or, the paper could simply drop the loose statements about how bad motifs are. A major conclusion of the cited Weirauch 2013 paper is actually that motif models aren't so bad, as long as you've got the right one - and the results in the present paper seem to support that idea.

(4) The new section analyzing the Y1H hit rate over motif matches as a function of predicted nucleosome occupancy is, in my opinion, one of the major novelties of the work - for the first time I think we are starting to get a good idea about the mechanisms underlying the false-negative rate in Y1H. However, both the explanation of the analyses and the interpretation of the data are hard to follow.

To begin, it is not explained exactly what is meant by PPV. Is this the fraction of Y1H hits that contain a motif? I think this is the case, since PPV is also stated to represent "Precision", and also because "Y1H sensitivity" is used as a separate measure (which I assume means the proportion of motifs that yield a Y1H hit). But it is never stated overtly.

I am also uncertain what is meant by "None" in the column "Nucleosome Occupancy Probability" in Table S5. Does this mean that the site is predicted to have no chance of nucleosome occupancy? Or that there is no filtering? I would think there would be virtually no sites with zero probability, so I'm going to ignore those rows. It would be very helpful if the Supplementary Tables had legends!

Assuming I am interpreting Table S5 correctly, and ignoring the "None" rows, then what I would take from these data is the following:

(A) As is noted in the paper, the B1H assay has much higher sensitivity if there is a "perfect" binding site, i.e. one that at $1E-5$ on FIMO. Suddenly, this notoriously insensitive assay has a sensitivity of 33%! This makes me think that sequence-specific interactions are only detected if the binding strength is very high. Has this ever been tested? i.e. insert sequences with known K_d , and ask whether the assay can detect them? If such a literature exists, it should be mentioned here what was the result. If it does not exist, the paper should perhaps mention that it would be beneficial to do such experiments.

(B) At this same motif score threshold, which is the only one that I can really take seriously (at least in this analysis) given the very low sensitivity at other thresholds (none of them exceeds 5%), there is a very low PPV. i.e. the vast majority of Y1H "hits" contain no motif. The paper later shows that at least some of these hits include sequences that do bind the protein, but I don't think the examples shown can account for the fact that it is such a small fraction. I suspect that the full-length TFs are being recruited indirectly by yeast factors that recognize the mouse regulatory sequences. Such phenomena appear to be common in metazoans. Perhaps the paper could speculate on whether this occurs in yeast as well?

(C) Table S5 appears to show that the sensitivity of the assay is greatly diminished if only the regions with high nucleosome occupancy are retained - there is a three-fold drop in sensitivity (33.8 to 11.8%) going from $P > 0.3$ to $P > 0.7$. This seems to indicate that nucleosomes may account for a majority of the variation in sensitivity in the Y1H assay! However, it is not clear exactly what is shown here. Are the occupancy scores the average over the entire regulatory region, or over the individual site? Are the counts done per site, or per regulatory region? And, why were the nucleosome occupancy categories $P > 0.3$, $P > 0.5$, $P > 0.7$, and none? More obvious categories would

be $0 < P < 0.25$, $0.25 < P < 0.5$, $0.5 < P < 0.75$, and $0.75 < P < 1.0$. This would reveal more explicitly how the probability of getting a "hit" relates to the predicted nucleosome occupancy over a site.

The paper, however, interprets the data in a very different way. In fact, the whole paragraph on this matter on P. 10 makes no sense to me. Here is my interpretation of this paragraph:

"All seven DNA baits showed an overall high nucleosome occupancy landscape. This suggests that many interactions, which involve sites that are predicted to be strongly occupied by nucleosomes are nevertheless detected by Y1H."

Essentially any piece of DNA will show an overall high nucleosome occupancy landscape, unless it is highly A-T rich or contains polyA/polyT stretches. Nucleosomes occupy the majority of eukaryotic genomes. Human regulatory regions are typically G-C enriched, making them especially likely to have high intrinsic nucleosome occupancy. But, not every base is highly occupied. What needs to be considered (and I think is considered here, although I'm not sure) is the predicted occupancy over the motif match.

"An example is the interaction between the Mcts2-Id1 enhancer and RFX2 whose motif is predicted to be located in a high nucleosome occupancy region ($P=0.88$)."

This is an interesting single observation, but I don't see how it is consistent with the overall trends in PPV. Also, maybe RFX2 has the ability to compete with nucleosomes? Some TFs do, although most don't. What is known about this one?

"This observation is consistent with the PPV values, which gradually decrease upon progressively excluding regions within DNA baits that feature high ($P > 0.7$), moderate to high ($P > 0.5$), or low to high ($P > 0.3$) predicted nucleosome occupancy probabilities (Supplementary Table S5). In other words, by removing regions within DNA baits with a moderate to high predicted nucleosome occupancy, we remove many motifs that support detected Y1H interactions. However, when calculating the corresponding sensitivity values, we observed an inverse relationship when considering all Y1H interactions supported by high stringency motifs."

It is actually not clear to me what is shown in Table S5, or why it is displayed this way - see comments above - but I am now guessing that "None" means no filtering? In any case, there is always a tradeoff between precision/specificity (which I think is what PPV means?) and recall (which is what Sensitivity means here). To achieve higher specificity/precision/PPV, you typically have to accept a lower recall value, and vice versa. This is well documented e.g. in the cancer screening literature and even in the origins of ROC analysis (radar detection). It is not stated how the analyses were done here exactly, but it seems quite likely that the higher PPV is simply because more DNA is included if you use lower motif scores and less stringent nucleosome occupancy cutoffs. Of course this has a disastrous impact on recall/sensitivity, as expected.

It is also not clear what proportion of all sequences meet the indicated criteria, and that would influence how the PPV values are interpreted. I somehow suspect that a motif cutoff of $1E-3$ and "None" for nucleosomes will place a large majority of sequences into the positive set (since, at random, there should be a weak motif hit every 1/1000 bases - see below), making it pretty easy to get a PPV of 89%. Perhaps what needs to be shown is the PPV divided by the expected PPV you would get by making random guesses.

"This suggests that sites that are predicted to be weakly occupied with nucleosomes and have a high motif score are more likely to be detected. Providing that the predicted data reflect true in vivo behavior, our analyses therefore do not point to nucleosome occupancy as the main factor that affects the sensitivity of our Y1H platform, although the most accessible consensus sites have a clearly greater detection likelihood."

I am baffled by this set of statements, which seems to be somewhere between wrong, misleading, and poorly written. There is no statistical analysis shown which indicates the relative importance of individual factors, so no claim can be made there; in any case, the analyses do support the importance of nucleosome occupancy, which is stated twice: the phrase "This suggests that sites that are predicted to be weakly occupied with nucleosomes and have a high motif score are more likely

to be detected" has the same meaning as "the most accessible consensus sites have a clearly greater detection likelihood", so they should be merged.

Overall, I applaud the inclusion of this analysis in the paper, but the presentation needs to be redone. I also don't see why the paragraph doesn't start with the basic observations - the whole thing is backwards. First, acknowledge that the assay only has high sensitivity when there is a perfect consensus sitting in a region of low nucleosome occupancy. Then, acknowledge that the assay has a very high false-positive rate with regard to motifs, under any situation where there is high sensitivity. Therefore, these hits must be due to something besides sequence-specific DNA-binding - perhaps speculate on what that might be.

Curiously, the next paragraph is much more sensible. I don't see why this one isn't merged with the previous paragraph that I find so problematic.

I also noticed in Figure S9 that the nucleosome occupancy decreases to zero at the borders of all of the DNA sequences analyzed. This presumably indicates that the insert sequences were plugged into the Kaplan model, but not the surrounding vector sequences. It is paramount to include the flanking sequence, since it is present in the vectors used in the assays, and will influence the nucleosome positions within the inserts substantially! Unfortunately I think this means that the analyses need to be rerun with the nucleosome predictions made on the full plasmid sequence, otherwise it is possible that the conclusions are erroneous. Alternatively, the analysis could be switched to a model more like the Tillo calculation, which doesn't account for adjacent sequences, it just gives a number for each base that is calculated from a surrounding nucleosome-sized window.

If this is done, it might also be worth scanning the sequences for binding sites for Reb1, Abf1, Rap1, Rsc3, Mcm1, and Tbf1. These are all yeast TFs that are capable of competing with and displacing nucleosomes. If there is a binding site for one of these, then it will probably kick off an overlapping nucleosome. This could explain the occasional site that binds a mouse TF despite having high intrinsic nucleosome occupancy.

(5) The section on luciferase testing needs to state how many of the novel interactions were positive in HEK293 cells - it's an obvious omission, and not giving the number suggests a shell game.

(6) The following in the discussion I believe is erroneous in its expectation - the numbers are actually fine: "This means that about 60 reactions are predicted per DNA bait when the latter are scanned for motifs using a default threshold. Indeed, it would imply that ~20% of all TFs within our current Y1H assay whose binding specificities have been characterized can interact with any given ~1,000 bp DNA sequence and that around 40% of each screened DNA sequence would be involved in TF-DNA interactions (i.e., based on motif scanning), which likely appears excessive".

In fact, this is almost exactly what would be expected from motif scans. Most human TFs have a binding site that is 6-9 bases wide, with degeneracy at multiple positions. Thus, they are about as specific as six-cutter restriction enzymes, in which sites occur at random every 4 kb - so, in a 1 kb fragment, we would expect binding sites for about 25% of all TFs. Wunderlich and Mirny (2009) reviewed this phenomenon. For most TFs, there is a vast excess of potential binding sites in the genome, and indeed there also appear to be an excess of bona fide binding sites that are "neutral" (i.e. not doing any gene regulation). Neither PWMs nor the system described in this paper deal with this problem, which is one of the major puzzles in regulatory genomics.

Reviewer #2 (Remarks to the Author):

The resubmission of Gubelman has addressed my primary concern, and incorporated most of my minor ones. I feel it is suitable for publication.

One minor point: I'd recommend re-inserting all the supplemental figures as higher-quality pngs or jpgs, as many were illegible (thus my "Low" scoring for the quality of supplemental info).

Reviewer 1:

Nevertheless, reviewer 1 requests to address a few outstanding questions or issues for the sake of increased clarity, which however “should not require re-review”. We have addressed these points below:

1a) The reviewer requests to better indicate how certain numbers in terms of positives in a screen or comparative analyses for example are obtained by pointing the reader to specific rows or columns in the respective Tables.

>Where appropriate, we have provided additional details as to the precise location of the discussed data in the Tables.

1b) The reviewer provides an example, indicating that “on P. 7, it is stated that “We identified 40 protein-DNA interactions involving 25 distinct TFs and seven DNA baits.” But, in the discussion of motifs, it says “In addition, 16 out of the 40 detected Y1H interactions involving 10 TFs were not predicted through motif scanning”. I cannot reconcile these numbers.”

>Both statements are correct, however, we appreciate that they may have caused confusion. We have therefore rephrased the sentence discussing the motif analyses as follows: “*In addition, 16 out of the 40 detected Y1H interactions were not predicted through motif scanning. These 16 interactions are mediated by 10 TFs. For two out of these 10 TFs,....*” (p.9 of the revised manuscript).

2) The reviewer wonders why PWMs for TFs that are not included in the Jolma et al. (Cell, 2013) dataset but that are available in other databases were not immediately included in the initial analysis pipeline.

>The generation of a single, curated, and comprehensive mouse TF PWM resource containing the optimal DNA binding site motif for each TF represents a significant challenge and constitutes a large research project by itself as has recently been illustrated for yeast TFs by the Bulyk group (Gordân et al., Genome Biology, 2011). To avoid PWM quality issues or discrepancies between methods and to streamline analyses, we have therefore selected to use a single mouse TF PWM data set, which is considered to be the most comprehensive and high-quality to date. We have in our opinion adequately motivated this choice in the manuscript when we indicate that: “*For consistency, we used a recently published high-throughput SELEX data set by Jolma et al. (2013) because it contains the largest collection of high-quality PWMs for human and mouse TFs to date, allowing us to bypass potential issues with PWM qualities which may vary dependent on the input data or method through which they were derived (Gordan et al, 2011; Medina-Rivera et al, 2011; Stormo & Zhao, 2010; Weirauch et al, 2013). We were able to retrieve reliable PWM data for 326 out of 750 TFs (43%) within our clone collection, based on the fact that these TFs were directly probed or at least their one-to-one human orthologues.*”

We then found 13 Y1H interactions involving eight TFs for which we did not have PWM data. A systematic exploration of other PWM databases subsequently revealed the availability of binding motif data for five of these eight TFs. By including this observation in the manuscript, we feel that we acknowledge the limitation of our initial analysis in that it did not consider all available TF PWM data. However, we believe that this entire analysis nevertheless allowed us to adequately address the interesting question of how our Y1H findings in general compare to those derived from *in silico* motif scanning. We would therefore be in favor of keeping the description of our findings as is.

3) The reviewer requests to clarify the following statement: “motif scanning may itself be prone to false positive calls (Klepper & Drablos, 2013; Medina-Rivera et al, 2011)” as the reviewer feels that these papers do not support “this statement beyond the obvious fact (already noted in my previous review) that, in vivo, not all motif matches are occupied for reasons such as nucleosome occupancy, competition among TFs, chromatin state, etc, even though the TF would bind the site in a biochemical experiment with purified DNA and protein. The Y1H assay is supposed to detect

protein-DNA interactions independently of chromatin (although clearly it does not really do that from the analyses presented here!) so the literature on motifs vs. DNA needs to be cited if the idea of motif models is going to be critiqued. I think the original papers from Bulyk, Stormo, Benos, etc and maybe some of the followup large-scale studies would probably be best - and it would be beneficial to the reader to know exactly why there would be false positive calls. Or, the paper could simply drop the loose statements about how bad motifs are. A major conclusion of the cited Weirauch 2013 paper is actually that motif models aren't so bad, as long as you've got the right one - and the results in the present paper seem to support that idea."

>We believe that our statement is not unfounded in that it is well acknowledged in the field that scanning algorithms are not perfect, which is what our statement portrays. Note that the reviewer indicates her- or himself that "a major conclusion of the cited Weirach 2013 paper is actually that the motif models aren't so bad" so this is more of a "glass half-full or half-empty" discussion. However, if one for example examines Figure 1 of this same paper, then we can see that in general, motif scanning performs well (as also indicated in the introduction of our manuscript), but that there are nevertheless several instances where binding events are predicted which do not correspond to detected binding events. This is consistent with our own MARE results which showed that the correlation between MARE-based DNA binding and predicted motif presence starts to deteriorate when considering low scoring motifs. Together, these observations suggest the occurrence of false positive motif scanning calls, which again, our statement simply implies. We do however acknowledge that it may be valuable to the reader to include in this phrase why we (and the field at large) believe that motif scans may produce false positives. We have therefore modified this sections as follows and also added additional references: "*These numbers reflect on the one hand the fact that motif scanning may itself be prone to false positive calls of which the rate depends on the type of algorithm, the quality of TF PWMs, or the scanning parameters (e.g. detection threshold) used (Gordan et al, 2011; Klepper & Drablos, 2013; Medina-Rivera et al, 2011; Weirauch et al, 2013), but on the other hand the likely substantial technical false negative rate of the YIH assay*" (p.10 of the revised manuscript).

4) The reviewer indicates that the "new section analyzing the YIH hit rate over motif matches as a function of predicted nucleosome occupancy is, in my opinion, one of the major novelties of the work - for the first time I think we are starting to get a good idea about the mechanisms underlying the false-negative rate in YIH. However, both the explanation of the analyses and the interpretation of the data are hard to follow."

4a. To begin, it is not explained exactly what is meant by PPV. Is this the fraction of YIH hits that contain a motif? I think this is the case, since PPV is also stated to represent "Precision", and also because "YIH sensitivity" is used as a separate measure (which I assume means the proportion of motifs that yield a YIH hit). But it is never stated overtly.

>The way the reviewer interprets the meaning of PPV and YIH sensitivity is correct. However, to eliminate any confusion, we have now included more elaborate explanations (than the one used so far) in the *Materials & Methods* section: "*To benchmark our YIH assay, we calculated the positive predictive value (PPV or precision rate), i.e., the fraction of YIH TF-DNA interactions that are supported by a motif hit, and sensitivity, i.e., the fraction of motif hits detected by YIH, for different YIH interaction sets (Supplementary Table S4)*" (p.30 of the revised manuscript).

4b. I am also uncertain what is meant by "None" in the column "Nucleosome Occupancy Probability" in Table S5. Does this mean that the site is predicted to have no chance of nucleosome occupancy? Or that there is no filtering? I would think there would be virtually no sites with zero probability, so I'm going to ignore those rows. It would be very helpful if the Supplementary Tables had legends!

>To avoid confusion, we relabeled "none" to "No filtering" and added a short explanation to the revised Supplementary Table S5. The PPV and sensitivity values in the "No filtering" column refer to the scenario in which no nucleosomes are considered for the PPV/sensitivity analysis.

4c. As is noted in the paper, the BIH assay has much higher sensitivity if there is a "perfect" binding site, i.e. one that at 1E-5 on FIMO. Suddenly, this notoriously insensitive assay has a sensitivity of 33%! This makes me think that sequence-specific interactions are only detected if the binding

strength is very high. Has this ever been tested? i.e. insert sequences with known Kd, and ask whether the assay can detect them? If such a literature exists, it should be mentioned here what was the result. If it does not exist, the paper should perhaps mention that it would be beneficial to do such experiments.

>Indeed, we state in our manuscript that “*This suggests that sites that are predicted to be weakly occupied by nucleosomes and have a high motif score are more likely to be detected with Y1H*”, consistent with the reviewer’s interpretation. The reviewer suggests an interesting experiment which, to our knowledge, has so far not been performed, but could indeed be interesting for a follow-up study, although we suspect that the outcome of such analyses may prove to be highly dependent on the type of TF that is studied.

4d. To explain the low PPV, the reviewer wonders whether Y1H interactions involving TFs for which we cannot find any predicted motifs in the target sequence despite having the respective PWM may be detected because the TFs are indirectly recruited to the target sequence through yeast TFs.

>Overall, the PPV is actually quite high, ranging from ~55% for Y1H consensus interactions at the most stringent detection threshold ($P=1E-5$) to ~91% at the lower-bound threshold ($P=1E-3$), indicating that most Y1H interactions are actually supported by a predicted motif. Nevertheless, we cannot exclude that some Y1H interactions are indirect in that they may be mediated by endogenous yeast TFs, thus possibly explaining why we cannot find any corresponding motifs in the target sequence. However, in the absence of any hard data in support of this hypothesis, we would prefer not to extensively speculate about this in the revised manuscript. Instead, we have slightly rephrased the following sentence by adding “yeast one-hybrid” to it: “*Yeast one-hybrid, motif scanning and luciferase reporter assays do not yield conclusive evidence that the tested TFs are directly binding to their respective DNA elements.*” (p.12 of the revised manuscript).

4e. Table S5 appears to show that the sensitivity of the assay is greatly diminished if only the regions with high nucleosome occupancy are retained there is a three-fold drop in sensitivity (33.8 to 11.8%) going from $P > 0.3$ to $P > 0.7$. This seems to indicate that nucleosomes may account for a majority of the variation in sensitivity in the Y1H assay! However, it is not clear exactly what is shown here. Are the occupancy scores the average over the entire regulatory region, or over the individual site? Are the counts done per site, or per regulatory region?

>We believe that the confusion in interpreting the data stems from the fact that the reviewer misunderstood the origin and meaning of the data in Supplementary Table S5. The Y1H sensitivity values namely correspond to motif predictions in DNA regions that are *not* covered by nucleosomes at a specific occupancy threshold. In other words, DNA positions with occupancy probabilities *above* a specific threshold were excluded (no averaging). However, the reviewer seems to have understood the converse, i.e. that the values correspond to motif predictions *within* DNA regions covered by nucleosomes. As also indicated above, we have now included additional explanations to both Supplementary Table S5 and the *Materials and Methods* section (p. 31 of the revised manuscript) and also changed the Table labeling to eliminate any possible confusion. Based on our data, we therefore stick to our original and intuitive interpretation that high affinity binding sites in regions with a low probability of nucleosome occupancy have a greater chance to be detected by Y1H.

4f. Essentially any piece of DNA will show an overall high nucleosome occupancy landscape, unless it is highly A-T rich or contains polyA/polyT stretches. Nucleosomes occupy the majority of eukaryotic genomes. Human regulatory regions are typically G-C enriched, making them especially likely to have high intrinsic nucleosome occupancy. But, not every base is highly occupied. What needs to be considered (and I think is considered here, although I'm not sure) is the predicted occupancy over the motif match.

>Our analyses considered that nucleosome occupancy probabilities are position-specific, i.e. if at least one residue of a binding site has a nucleosome probability above the threshold, then this site will no longer be considered to be detectable by Y1H. We have included this in the respective *Materials and Methods* section (p.31) of the revised manuscript.

4g. *"An example is the interaction between the Mcts2-Id1 enhancer and RFX2 whose motif is predicted to be located in a high nucleosome occupancy region ($P=0.88$)."* This is an interesting single observation, but I don't see how it is consistent with the overall trends in PPV. Also, maybe RFX2 has the ability to compete with nucleosomes? Some TFs do, although most don't. What is known about this one?

>Again, we believe that the reviewer's confusion stems from the misinterpretation of Supplementary Table S5. Given the additional explanations provided above and also in the revised manuscript, we therefore believe that the statement in our original manuscript is correct: *"An example is the interaction between the Mcts2-Id1 enhancer and RFX2 whose motif is predicted to be located in a high nucleosome occupancy region ($P=0.88$). This observation is consistent with the PPV values, which gradually decrease upon progressively excluding regions within DNA baits that feature high ($P>0.7$), moderate to high ($P>0.5$), or low to high ($P>0.3$) predicted nucleosome occupancy probabilities (Supplementary Table S5). In other words, by removing regions within DNA baits with a moderate to high predicted nucleosome occupancy, we remove many motifs that support detected YIH interactions"* (p.10 of the revised manuscript). Thus, *"this suggests that many interactions, which involve sites that are predicted to be strongly occupied by nucleosomes are nevertheless detected by YIH"* (p.10 of the revised manuscript). We could not find any reports on the DNA binding behavior of RFX2 with regards to nucleosome competition consistent with it being a poorly characterized TF in general.

4h. *It is also not clear what proportion of all sequences meet the indicated criteria, and how that would influence how the PPV values are interpreted.*

→ For the 7 DNA baits:

- Weak-to-strong: 98% of the sequences are covered by nucleosomes
- Moderate-to-strong: 90% of the sequences are covered by nucleosomes
- Strong: 62% of the sequences are covered by nucleosomes

We have included this info in the manuscript (revised Supplementary Table S5). Since an increasing amount of sequence gets 'excluded' as we lower the nucleosome occupancy threshold, it is expected that motifs will increasingly be lost, which in turn will lower the PPV providing that these motifs are indeed located within the excluded sequences. Thus, this indicates that many YIH interactions occur in DNA regions predicted to have nucleosomes as also explained above.

4i. *I somehow suspect that a motif cutoff of $1E-3$ and "None" for nucleosomes will place a large majority of sequences into the positive set (since, at random, there should be a weak motif hit every 1/1000 bases - see below), making it pretty easy to get a PPV of 89%. Perhaps what needs to be shown is the PPV divided by the expected PPV you would get by making random guesses.*

>The reviewer makes a valid point, however, we would like to point out that the FIMO motif prediction algorithm already takes into account the probability over background to report a motif hit.

4j. *"This suggests that sites that are predicted to be weakly occupied with nucleosomes and have a high motif score are more likely to be detected. Providing that the predicted data reflect true in vivo behavior, our analyses therefore do not point to nucleosome occupancy as the main factor that affects the sensitivity of our YIH platform, although the most accessible consensus sites have a clearly greater detection likelihood." I am baffled by this set of statements, which seems to be somewhere between wrong, misleading, and poorly written. There is no statistical analysis shown which indicates the relative importance of individual factors, so no claim can be made there; in any case, the analyses do support the importance of nucleosome occupancy, which is stated twice: the phrase "This suggests that sites that are predicted to be weakly occupied with nucleosomes and have a high motif score are more likely to be detected" has the same meaning as "the most accessible consensus sites have a clearly greater detection likelihood", so they should be merged.*

>We hope that by clarifying the meaning of the data in Supplementary Table S5, our interpretation has become more intuitive. On the one hand, we show that many regions with high nucleosome occupancy are involved in interactions identified by YIH, suggesting that nucleosomes do not appear to interfere with their detection. On the other hand, we found that high affinity sites that are located in regions that are weakly occupied by nucleosomes have a greater probability of being detected. Together, these observations suggest that nucleosomes indeed likely impact the detection performance of our YIH assay, although they do not always prevent interactions from taking place

and thus from being detected. We have now rephrased this section in the revised manuscript accordingly (p.11): *“Providing that the predicted data reflect true in vivo behavior, our analyses point to nucleosome occupancy as another plausible factor affecting the predictive value of our YIH platform.”* We note that this statement is consistent with the one in the discussion of the original (and also revised) manuscript (p.19): *“Indeed, there are several reasons why the YIH assay may fail to detect specific protein-DNA interactions. For example, we showed that the nucleosome occupancy landscape, the number of TF binding sites per DNA bait, and score of motifs within DNA baits may affect the detection probability.”*

4k. I also noticed in Figure S9 that the nucleosome occupancy decreases to zero at the borders of all of the DNA sequences analyzed. This presumably indicates that the insert sequences were plugged into the Kaplan model, but not the surrounding vector sequences. It is paramount to include the flanking sequence, since it is present in the vectors used in the assays, and will influence the nucleosome positions within the inserts substantially! Unfortunately I think this means that the analyses need to be rerun with the nucleosome predictions made on the full plasmid sequence, otherwise it is possible that the conclusions are erroneous. Alternatively, the analysis could be switched to a model more like the Tillo calculation, which doesn't account for adjacent sequences, it just gives a number for each base that is calculated from a surrounding nucleosome-sized window.

>We thank the reviewer for pointing out this issue. In response, we have now included the vector flanking sequences (1 kb on each side) to avoid any boundary effects. The general patterns and interpretations were however not altered by this re-analysis.

4l. If this is done, it might also be worth scanning the sequences for binding sites for Reb1, Abf1, Rap1, Rsc3, Mcm1, and Tbf1. These are all yeast TFs that are capable of competing with and displacing nucleosomes. If there is a binding site for one of these, then it will probably kick off an overlapping nucleosome. This could explain the occasional site that binds a mouse TF despite having high intrinsic nucleosome occupancy.

>This is an interesting argument, although a highly speculative one, especially given the fact that the calculated nucleosome occupancies are only predictions themselves. We therefore believe that this issue should be addressed experimentally in a follow-up project as this would ideally involve the screening of regulatory elements in yeast cells in which single yeast TFs would have been mutated after which the detected interactions would be compared to those obtained with screening the same elements in corresponding “wildtype” cells.

5) The section on luciferase testing needs to state how many of the novel interactions were positive in HEK293 cells - it's an obvious omission, and not giving the number suggests a shell game.

>We thank the reviewer to point out this omission. We have corrected this in the revised manuscript (p.11): *“Furthermore, out of 33 novel YIH-detected interactions, 26 (79%) were positive in HEK293 cells.”*

6) The following in the discussion I believe is erroneous in its expectation - the numbers are actually fine: "This means that about 60 reactions are predicted per DNA bait when the latter are scanned for motifs using a default threshold. Indeed, it would imply that ~20% of all TFs within our current YIH assay whose binding specificities have been characterized can interact with any given ~1,000 bp DNA sequence and that around 40% of each screened DNA sequence would be involved in TF-DNA interactions (i.e., based on motif scanning), which likely appears excessive". In fact, this is almost exactly what would be expected from motif scans. Most human TFs have a binding site that is 6-9 bases wide, with degeneracy at multiple positions. Thus, they are about as specific as six-cutter restriction enzymes, in which sites occur at random every 4 kb - so, in a 1 kb fragment, we would expect binding sites for about 25% of all TFs. Wunderlich and Mirny (2009) reviewed this phenomenon. For most TFs, there is a vast excess of potential binding sites in the genome, and indeed there also appear to be an excess of bona fide binding sites that are "neutral" (i.e. not doing any gene regulation). Neither PWMs nor the system described in this paper deal with this problem, which is one of the major puzzles in regulatory genomics.

>We concur with the reviewer's reasoning, but nevertheless would argue that, consistent with our arguments raised above, motif scanning also generates false positive calls (e.g., due to ambiguity of PWMs, motif scanning algorithms, motif scanning detection thresholds, background models). These

would obviously inflate the false negative rate of our Y1H assay, which is what we actually meant with the quoted statement. However, in light of this discussion, we have now rephrased this section to: *“While the Y1H assay has likely a substantial false negative rate (see also below), it is unlikely that all of these predicted interactions constitute true positives due to ambiguity in TF PWMs, motif scanning algorithms, motif scanning detection thresholds, and background models.”*

Reviewer 2:

One minor point: I'd recommend re-inserting all the supplemental figures as higher-quality pngs or jpgs, as many were illegible (thus my "Low" scoring for the quality of supplemental info).

>We are not sure which Supplementary Figures in particular the reviewer is referring to, because they appear to be fine in our PDF file and so perhaps something went wrong during the upload? It is true that some of the Y1H assay pictures are of lower resolution, but this is something that unfortunately cannot be improved upon. Nevertheless, we will work diligently with the copy editors to make sure that all Supplementary Figures will be clear and legible.

Editor:

It seems also that error bars are provided in several figures even if the number of replicates is only of two (Fig 3B, 6A, supp fig S17, for example) and that technical and biological replicate are all considered as independent samples. For example the legend of Figure 3AB states "The error bars represent the standard error of two independent experiments with three replicates each (n=6)". Please correct the analysis and data presentation to follow our guidelines (see also <http://www.nature.com/msb/authors/index.html#a3.4.3>) : "Graphs must include clearly labelled error bars for cases where more than two independent experiments have been performed (error bars should not be shown for technical replicates)." and "For each experiment, the number of both technical and biological replicates should be clearly stated. Biological replicates are derived from independent experiments using separately obtained biological samples, while technical replicates are created by repeated measurements on the same biological sample. In general, technical replicates should be averaged before any statistical inference tests are performed". For these figures (reporter assays), please upload the Excel files (or csv) that include the individual values for technical and biological replicates as 'figure source data' associated with the respective panels (see also 'Source data for figures' under <http://www.nature.com/msb/authors/index.html#a3.4.3>).

>Our dual luciferase reporter data are in fact based on six (Figure 3A, B) or ten (Figure 6B) independent biological replicates, i.e. each independent replicate constitutes a well of cells that were independently transfected with the reporter constructs, lysed, and prepared for measurement. We have now clearly indicated this in the revised manuscript in the *Materials and Methods* section (p.28). We have also included the source data for the respective Figures as part of the resubmission.